# Mitigating Over-smoothing in Transformers via Regularized Nonlocal Functionals

**Tam Nguyen**
Department of Electrical & Computer Engineering
Rice University
Houston, USA
mn72@rice.edu

**Tan M. Nguyen**
Department of Mathematics
National University of Singapore
Singapore
tanmn@nus.edu.sg

**Richard G. Baraniuk**
Department of Electrical & Computer Engineering
Rice University
Houston, USA
richb@rice.edu

## Abstract

Transformers have achieved remarkable success in a wide range of natural language processing and computer vision applications. However, the representation capacity of a deep transformer model is degraded due to the over-smoothing issue in which the token representations become identical when the model's depth grows. In this work, we show that self-attention layers in transformers minimize a functional which promotes smoothness, thereby causing token uniformity. We then propose a novel regularizer that penalizes the norm of the difference between the smooth output tokens from self-attention and the input tokens to preserve the fidelity of the tokens. Minimizing the resulting regularized energy functional, we derive the Neural Transformer with a Regularized Nonlocal Functional (NeuTRENO), a novel class of transformer models that can mitigate the over-smoothing issue. We empirically demonstrate the advantages of NeuTRENO over the baseline transformers and state-of-the-art methods in reducing the over-smoothing of token representations on various practical tasks, including object classification, image segmentation, and language modeling.

## 1 Introduction

Transformer models [62] have achieved substantial success in natural language processing [16, 2, 13, 10, 47, 4, 6, 14], reinforcement learning [9, 32], computer vision [19, 40, 59, 49, 44, 3, 41, 71, 27], and other practical applications [50, 33, 70, 26, 66]. Transformers also excel at transferring knowledge from pre-trained models to new tasks, even when limited supervision is available [45, 46, 16, 69, 39]. At the heart of transformers lies the self-attention mechanism, which computes a weighted average of token representations within a sequence. These weights are determined based on the similarity scores between pairs of tokens, determining their relative importance in the sequence [11, 43, 38]. This flexibility in capturing diverse syntactic and semantic relationships has been identified as a crucial factor contributing to the success of transformers [57, 63, 12, 64, 31].

### 1.1 Background: Self-Attention

For a given input sequence $\mathbf{X} := [\boldsymbol{x}(1), \cdots, \boldsymbol{x}(N)]^\top \in \mathbb{R}^{N \times D_x}$ of $N$ feature vectors, self-attention transforms $\mathbf{X}$ into the output sequence $\mathbf{H}$ in the following two steps:

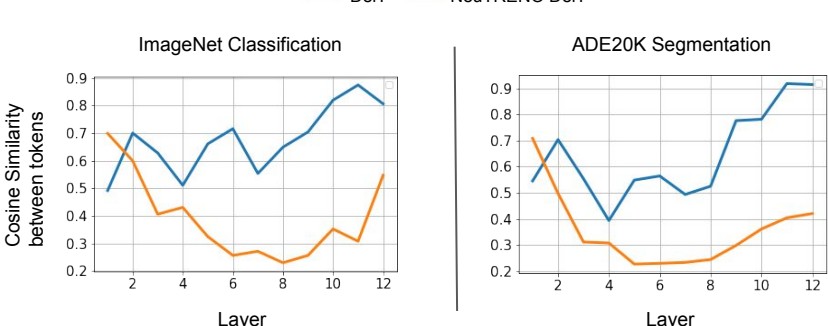

Figure 1: The cosine similarity between tokens representations across layers of NeuTRENO DeiT vs. the baseline DeiT models on the Imagenet classification and ADE20K image segmentation tasks. In both tasks, the DeiT baseline suffers from over-smoothing as tokens become similar to identical when the model gets deeper. In contrast, tokens in NeuTRENO models are significantly more diverse, suggesting a reduction in over-smoothing. Further details regarding this analysis can be found in Appendix E.

**Step 1.** The input sequence $\mathbf{X}$ is projected into the query matrix $\mathbf{Q}$, the key matrix $\mathbf{K}$, and the value matrix $\mathbf{V}$ via three linear transformations

$$\mathbf{Q} = \mathbf{X}\mathbf{W}_Q^\top; \mathbf{K} = \mathbf{X}\mathbf{W}_K^\top; \mathbf{V} = \mathbf{X}\mathbf{W}_V^\top, \tag{1}$$

where $\mathbf{W}_Q, \mathbf{W}_K \in \mathbb{R}^{D_{qk} \times D_x}$, and $\mathbf{W}_V \in \mathbb{R}^{D \times D_x}$ are the weight matrices. We denote $\boldsymbol{Q} := [\boldsymbol{q}(1), \dots, \boldsymbol{q}(N)]^\top, \mathbf{K} := [\boldsymbol{k}(1), \dots, \boldsymbol{k}(N)]^\top$, and $\mathbf{V} := [\boldsymbol{v}(1), \dots, \boldsymbol{v}(N)]^\top$, where the vectors $\boldsymbol{q}(i), \boldsymbol{k}(i)$, and $\boldsymbol{v}(i)$, for $i = 1, \dots, N$ are the query, key, and value vectors, respectively.

**Step 2.** The output sequence $\mathbf{U} := [\boldsymbol{u}(1), \dots, \boldsymbol{u}(N)]^\top \in \mathbb{R}^{N \times D_{qk}}$ is then computed as follows

$$\mathbf{U} = \mathrm{softmax}\Big(\mathbf{Q}\mathbf{K}^\top/\sqrt{D_{qk}}\Big)\mathbf{V} := \mathbf{A}\mathbf{V}, \tag{2}$$

where the softmax function is applied to each row of the matrix $\mathbf{Q}\mathbf{K}^\top/\sqrt{D_{qk}}$. The matrix $\mathbf{A} := \mathrm{softmax}\Big(\frac{\mathbf{Q}\mathbf{K}^\top}{\sqrt{D_{qk}}}\Big) \in \mathbb{R}^{N \times N}$ and its component $a_{ij}$ for $i, j = 1, \cdots, N$ are called the attention matrix and attention scores, respectively. For each query vector $\boldsymbol{q}(i)$ for $i = 1, \cdots, N$, an equivalent form of Eqn. (2) to compute the output vector $\boldsymbol{u}(i)$ is given by

$$\boldsymbol{u}(i) = \sum_{j=1}^{N} \mathrm{softmax}\Big(\boldsymbol{q}(i)^\top \boldsymbol{k}(j)/\sqrt{D_{qk}}\Big)\boldsymbol{v}(j). \tag{3}$$

The self-attention computed by Eqn. (2) and (3) is refered as softmax attention. In our work, we refer to a transformer that uses softmax attention as a softmax transformer.

## 1.2 Over-smoothing in Transformers

Despite their remarkable success, deep transformer-based models have been observed to suffer from the over-smoothing issue, in which all token representations become identical when more layers are added to the models [55, 65, 18]. This over-smoothing phenomenon, also known as the "token uniformity" problem, significantly limits the representation capacity of transformers. To illustrate this phenomenon, we examine the average cosine similarity between pairs of token representations across different layers in a softmax transformer trained for the Imagenet object classification and ADK20 image segmentation tasks [73]. As depicted in Fig. 1, in both tasks, this cosine similarity between tokens increases as the models become deeper. Particularly, in the last two layers, the cosine similarity scores are approximately 0.9, indicating a high degree of similarity among tokens.

## 1.3 Contribution

We develop a nonlocal variational denoising framework for self-attention, providing insights into the over-smoothing phenomenon in transformers. In particular, by viewing self-attention as a gradient descent step toward minimizing a nonlocal functional that penalizes high-frequency noise in the signal, we uncover the diffusive nature of self-attention, which explains the over-smoothing issue of transformers. Motivated by this understanding, we propose the Neural Transformer with a Regularized

Nonlocal Functional (NeuTRENO), a novel class of transformers designed to mitigate over-smoothing. NeuTRENO is derived by optimizing a regularized nonlocal functional, which includes an additional convex fidelity term. This fidelity term penalizes the norm of the difference between the smooth output tokens from self-attention and the input tokens, thereby reducing the over-smoothing effect. Our contribution is three-fold.

1. We develop a nonlocal variational denoising framework for self-attention and shed light on the over-smoothing issue that hampers the representation capacity of transformers.

2. We develop NeuTRENO, a novel class of transformers that are capable of alleviating the over-smoothing issue.

3. We theoretically prove that transformers with softmax self-attention are prone to over-smoothing while NeuTRENO can avoid this issue.

We empirically demonstrate the benefits of NeuTRENO on various large-scale applications, including the ImageNet object classification, ADE20K image segmentation, and WikiText-103 language modeling tasks.

**Organization**: We organize our paper as follows: in Section 2, we develop a nonlocal variational denoising framework for self-attention and provide an explanation for the over-smoothing issue in transformer-based models. In section 3, we propose NeuTRENO, and present a theoretical result that guarantees NeuTRENO's capability of mitigating over-smoothing. In Section 4, we empirically validate the benefits of NeuTRENO. We discuss the related work in Section 6. Finally, we conclude our main contributions and remarks. Further results, details, and proofs are provided in the Appendix.

## 2 A Nonlocal Variational Denoising Framework for Self-attention

We first consider the output matrix $\mathbf{U} := [\boldsymbol{u}(1), \cdots, \boldsymbol{u}(N)]^\top \in \mathbb{R}^{N \times D}$ in self-attention as given by Eqn. 2 in Section 1.1. Let $\Omega \subset \mathbb{R}$, $x \in \Omega$, and $\boldsymbol{u}(x) := [u_1(x), \ldots, u_D(x)]^T$ be a real vector-valued function, $\boldsymbol{u} : \Omega \to \mathbb{R}^D$, $\boldsymbol{u} \in L^2(\Omega)$. The output matrix $\mathbf{U}$ in self-attention discretizes the function $\boldsymbol{u}(x)$ on a 1-D grid. In the context of signal/image denoising, $\mathbf{U}$ can be considered as the *desired clean signal*, and $\boldsymbol{u}(x)$ is its corresponding intensity function denoting the signal values at the position $x \in \Omega$. We further let the observed intensity function $\boldsymbol{f}(x)$ denote the values of the *observed noisy signal* at $x \in \Omega$, $\boldsymbol{f} : \Omega \to \mathbb{R}^D$, $\boldsymbol{f} \in L^2(\Omega)$. For example, $\boldsymbol{f}(x)$ can be given as

$$\boldsymbol{f}(x) = \boldsymbol{u}(x) + \boldsymbol{n}(x), \tag{4}$$

where $\boldsymbol{n}$ is the additive noise. We wish to reconstruct $\boldsymbol{u}(x)$ from $\boldsymbol{f}(x)$. Following the variational denoising method proposed in [23] and [24], the denoised image $\boldsymbol{u}(x)$ can be obtained by minimizing the following regularized functional with respect to $\boldsymbol{u}$:

$$E(\boldsymbol{u}, \boldsymbol{f}) = J(\boldsymbol{u}) + G(\boldsymbol{u}, \boldsymbol{f}) \tag{5}$$
$$= \frac{1}{2} \int_{\Omega \times \Omega} \|\boldsymbol{u}(x) - \boldsymbol{u}(y)\|_2^2 k(x, y) dx dy + \frac{\lambda}{2} \int_{\Omega} \|\boldsymbol{u}(x) - \boldsymbol{f}(x)\|_2^2 dx.$$

Here, $J(\boldsymbol{u}) = \frac{1}{2} \int_{\Omega \times \Omega} \|\boldsymbol{u}(x) - \boldsymbol{u}(y)\|_2^2 k(x, y) dx dy$ is a nonlocal functional of weighted differences. The weights $k(x, y)$ represent the affinity between signal values at positions $x$ and $y$. For example, for images, $k(x, y)$ captures the proximity between pixels $x$ and $y$ in the image. $J(\boldsymbol{u})$ works as a regularizer. Minimizing $J(\boldsymbol{u})$ promotes the smoothness of $\boldsymbol{u}$ and penalizes high-frequency noise in the signal. Adding the convex fidelity term $G(\boldsymbol{u}, \boldsymbol{f}) = \frac{\lambda}{2} \int_{\Omega} \|\boldsymbol{u}(x) - \boldsymbol{f}(x)\|_2^2 dx$ to the functional $J(\boldsymbol{u})$ allows the denoised signal $\boldsymbol{u}(x)$ to preserve relevant information in the observed noisy signal $\boldsymbol{f}(x)$. The regularized functional $E(\boldsymbol{u}, \boldsymbol{f})$ can be considered as an energy functional.

### 2.1 Self-attention as a Gradient Descent Step to Minimize the Nonlocal Functional $J$

We show that self-attention is equivalent to taking a gradient descent step toward minimizing the functional $J(\boldsymbol{u})$ in the energy functional $E(\boldsymbol{u}, \boldsymbol{f})$. We expand $J(\boldsymbol{u})$ as follows

$$J(\boldsymbol{u}) = \frac{1}{2} \int_{\Omega \times \Omega} \sum_{j=1}^{D} (u_j(x) - u_j(y))^2 k(x, y) dx dy \tag{6}$$

The gradient of $J$ with respect to $\boldsymbol{u}$ is then given by

$$\nabla_{\boldsymbol{u}} J(\boldsymbol{u}) = \left[ \frac{\partial J}{\partial u_1}, \frac{\partial J}{\partial u_2}, \ldots, \frac{\partial J}{\partial u_D} \right]^T. \tag{7}$$

The partial derivative $\partial J/\partial u_j$, $j = 1, 2, \ldots, D$, is defined through its dot product with an arbitrary function $h_j \in L^2(\Omega)$ as follows

$$
\begin{aligned}
\frac{\partial J}{\partial u_j} \cdot h_j(x) &= \frac{d}{d\tau} J(u_j + \tau h_j)\big|_{\tau=0} \\
&= \frac{1}{2}\left(\frac{d}{d\tau}\int_{\Omega\times\Omega}(u_j(x) - u_j(y) + \tau h_j(x) - \tau h_j(y))^2 k(x,y)dxdy\right)\bigg|_{\tau=0} \\
&= \left(\int_{\Omega\times\Omega}(u_j(x) - u_j(y) + \tau h_j(x) - \tau h_j(y))(h_j(x) - h_j(y))k(x,y)dxdy\right)\bigg|_{\tau=0} \\
&= \int_{\Omega\times\Omega}(u_j(x) - u_j(y))(h_j(x) - h_j(y))k(x,y)dxdy \\
&= \int_{\Omega\times\Omega}(u_j(x) - u_j(y))h_j(x)k(x,y)dxdy - \int_{\Omega\times\Omega}(u_j(x) - u_j(y))h_j(y)k(x,y)dxdy
\end{aligned}
$$

Applying a change of variables $(x, y) \to (y, x)$ to the second term of the above integral, we have

$$
\begin{aligned}
\frac{\partial J}{\partial u_j} \cdot h_j(x) &= \int_{\Omega\times\Omega}(u_j(x) - u_j(y))h_j(x)k(x,y)dxdy - \int_{\Omega\times\Omega}(u_j(y) - u_j(x))h_j(x)k(y,x)dxdy \\
&= \int_{\Omega\times\Omega}(u_j(x) - u_j(y)(k(x,y) + k(y,x))dy h_j(x)dx
\end{aligned}
$$

Thus, the Frechet derivative of J with respect to $u_j$ is given by

$$
\frac{\partial J}{\partial u_j} = \int_{\Omega}(u_j(x) - u_j(y)(k(x,y) + k(y,x))dy. \tag{8}
$$

Substituting the formula for $\partial J/\partial u_j$ in Eqn. 8 into Eqn. 7 for $\nabla_{\boldsymbol{u}}J(\boldsymbol{u})(x)$, we obtain the following gradient flow

$$
\frac{d\boldsymbol{u}(x,t)}{dt} = -\nabla_{\boldsymbol{u}}J(\boldsymbol{u}) = \int_{\Omega}\big(\boldsymbol{u}(y,t) - \boldsymbol{u}(x,t)\big)\big(k(x,y) + k(y,x)\big)dy, \tag{9}
$$

where $t$ is the time variable we introduce to capture the dynamics of $\boldsymbol{u}$ when gradient descent is applied to minimize $J(\boldsymbol{u})$. Let $\boldsymbol{v}(x) := [v_1(x), \ldots, v_D(x)]^T$ be a real vector-valued function, $\boldsymbol{v}: \Omega \to \mathbb{R}^D$, $\boldsymbol{v} \in L^2(\Omega)$. We discretize $\boldsymbol{v}(x)$ on a 1-D grid to attain the value vectors $\boldsymbol{v}(1), \ldots, \boldsymbol{v}(N) \in \mathbb{R}^D$, which form the value matrix $\mathbf{V} := [\boldsymbol{v}(1), \cdots, \boldsymbol{v}(N)]^\top \in \mathbb{R}^{N\times D}$ in self-attention as defined in Eqn. 2. We initialize $\boldsymbol{u}$ at $t = 0$ with $\boldsymbol{v}(x)$, i.e., $\boldsymbol{u}(x,0) = \boldsymbol{v}(x)$.

**Self-attention is an Euler Discretization of the Gradient Flow Given in 9.** We discretize the gradient flow in Eqn. 9 using the Euler method [21] with step size $\Delta t(x) = 1/\int_{\Omega}\big(k(x,y) + k(y,x)\big)dy$ and obtain the following update

$$
\begin{aligned}
\boldsymbol{u}(x, \Delta t(x)) &= \boldsymbol{u}(x,0) + \Delta t(x)\int_{\Omega}\big(\boldsymbol{u}(y,0) - \boldsymbol{u}(x,0)\big)\big(k(x,y) + k(y,x)\big)dy \\
&= \int_{\Omega}\frac{\big(k(x,y) + k(y,x)\big)\boldsymbol{u}(y,0)}{\int_{\Omega}\big(k(x,y') + k(y',x)\big)dy'}dy = \int_{\Omega}\frac{K(x,y)\boldsymbol{v}(y)}{\int_{\Omega}K(x,y')dy'}dy. 
\end{aligned}\tag{10}
$$

Here, $K(x,y) := k(x,y) + k(y,x)$ is a symmetric kernel and $\boldsymbol{u}(y,0) = \boldsymbol{v}(y)$ since $\boldsymbol{u}$ is initialized at $t = 0$ with $\boldsymbol{v}$ as aforementioned. Let $\boldsymbol{k}(x) := [k_1(x), \ldots, k_{D_{qk}}(x)]^T$ be a real vector-valued function, $\boldsymbol{k}: \Omega \to \mathbb{R}^{D_{qk}}$, $\boldsymbol{k} \in L^2(\Omega)$. Similar to $\boldsymbol{u}(x)$ and $\boldsymbol{v}(x)$, we can discretize $\boldsymbol{k}(x)$ on a 1-D grid to attain the key vectors $\boldsymbol{k}(1), \ldots, \boldsymbol{k}(N) \in \mathbb{R}^{D_{qk}}$, which form the key matrix $\mathbf{K} := [\boldsymbol{k}(1), \cdots, \boldsymbol{k}(N)]^\top \in \mathbb{R}^{N\times D_{qk}}$ in self-attention as defined in Eqn. 2. We choose $K(x,y) = \exp\big(\boldsymbol{k}(x)^T\boldsymbol{k}(y)/\sqrt{D_{qk}}\big)$ and rewrite Eqn. 10 as follows

$$
\boldsymbol{u}(x, \Delta t(x)) = \int_{\Omega}\frac{\exp\big(\boldsymbol{k}(x)^T\boldsymbol{k}(y)/\sqrt{D_{qk}}\big)}{\int_{\Omega}\exp\big(\boldsymbol{k}(x)^T\boldsymbol{k}(y')/\sqrt{D_{qk}}\big)dy'}\boldsymbol{v}(y)dy. \tag{11}
$$

Estimating the integrals in Eqn. 11 via Monte-Carlo approximation using the key vectors $\boldsymbol{k}(1), \ldots, \boldsymbol{k}(N) \in \mathbb{R}^{D_{qk}}$ and and value vectors $\boldsymbol{v}(1), \ldots, \boldsymbol{v}(N) \in \mathbb{R}^D$, we obtain

$$\boldsymbol{u}(x, \Delta t(x)) \approx \sum_{j=1}^{N} \frac{\exp\big(\boldsymbol{k}(x)^T \boldsymbol{k}(j)/\sqrt{D_{qk}}\big)}{\sum_{j'=1}^{N} \exp\big(\boldsymbol{k}(x)^T \boldsymbol{k}(j')/\sqrt{D_{qk}}\big)} \boldsymbol{v}(j). \tag{12}$$

Discretizing $\boldsymbol{u}(x, \Delta t(x))$ on another 1-D grid, we attain

$$\boldsymbol{u}(i) \approx \sum_{j=1}^{N} \frac{\exp\big(\boldsymbol{k}(i)^T \boldsymbol{k}(j)/\sqrt{D_{qk}}\big)}{\sum_{j'=1}^{N} \exp\big(\boldsymbol{k}(i)^T \boldsymbol{k}(j')/\sqrt{D_{qk}}\big)} \boldsymbol{v}(j)$$

$$= \sum_{j=1}^{N} \operatorname{softmax}\Big(\boldsymbol{k}(i)^\top \boldsymbol{k}(j)/\sqrt{D_{qk}}\Big) \boldsymbol{v}(j), \quad i = 1, \ldots, N. \tag{13}$$

Comparing Eqn. 13 and Eqn. 3, we observe that Eqn. 13 implement a symmetric self-attention, in which the query matrix $\mathbf{Q}$ and the key matrix $\mathbf{K}$ are the same, i.e. $\mathbf{W}_Q = \mathbf{W}_K$ where $\mathbf{W}_Q$ and $\mathbf{W}_K$ are the linear projections that map the input sequence $\mathbf{X}$ into $\mathbf{Q}$ and $\mathbf{K}$ as given in Eqn. 1. This symmetry of the attention scores is desirable in some image processing tasks due to the symmetric similarities between pixels, but can be relaxed for other tasks. To break the symmetry of attention scores in Eqn. 13, we replace the key vectors $\boldsymbol{k}(i)$ by the query vectors $\boldsymbol{q}(i)$, $i = 1, \ldots, N$, to obtain the exact formula of self-attention given by Eqn. 3. The following theorem summarizes our results:

**Theorem 1** (Self-attention as a Gradient Descent Step to Minimize a Nonlocal Functional). *Given the nonlocal functional $J(\boldsymbol{u}) = \frac{1}{2} \int_{\Omega \times \Omega} \|\boldsymbol{u}(x) - \boldsymbol{u}(y)\|_2^2 k(x, y) dx dy$ of a vector-valued function $\boldsymbol{u} : \Omega \to \mathbb{R}^D$, $\boldsymbol{u} \in L^2(\Omega)$, and let $K(x, y) := k(x, y) + k(y, x) = \exp\big(\boldsymbol{k}(x)^T \boldsymbol{k}(y)/\sqrt{D_{qk}}\big)$, where $\boldsymbol{k} : \Omega \to \mathbb{R}^{D_{qk}}$, $\boldsymbol{k} \in L^2(\Omega)$. Then, taking a gradient descent step on $\boldsymbol{u}$ at time $t = 0$, where $\boldsymbol{u}(x, 0) = \boldsymbol{v}(x)$, with an adaptive step size $\Delta t(x) := \dfrac{1}{\int_{\Omega} \big(k(x, y) + k(y, x)\big) dy}$ to minimize $J$ is equivalent to updating $\boldsymbol{u}$ via a symmetric self-attention*

$$\boldsymbol{u}(x, \Delta t(x)) = \sum_{j=1}^{N} \operatorname{softmax}\Big(\boldsymbol{k}(x)^\top \boldsymbol{k}(j)/\sqrt{D_{qk}}\Big) \boldsymbol{v}(j),$$

*which results in*

$$\boldsymbol{u}(i) = \sum_{j=1}^{N} \operatorname{softmax}\Big(\boldsymbol{k}(i)^\top \boldsymbol{k}(j)/\sqrt{D_{qk}}\Big) \boldsymbol{v}(j), \quad i = 1, \ldots, N. \tag{14}$$

*Here, $\boldsymbol{u}(n)$, $\boldsymbol{v}(n)$, and $\boldsymbol{u}(n)$, $n = 1, \ldots, N$, are the key, value, and output vectors in self-attention, respectively. Breaking the symmetry of the attention scores by replacing $\boldsymbol{k}(i)$ with $\boldsymbol{q}(i)$, $i = 1, \ldots, N$, in Eqn. 14, we obtain the exact formula of self-attention*

$$\boldsymbol{u}(i) = \sum_{j=1}^{N} \operatorname{softmax}\Big(\boldsymbol{q}(i)^\top \boldsymbol{k}(j)/\sqrt{D_{qk}}\Big) \boldsymbol{v}(j), \quad i = 1, \ldots, N.$$

**Remark 1.** *In Eqn. 9, the change in $\boldsymbol{u}$ at position $x$ is proportional to the sum of differences between $\boldsymbol{u}(x)$ and $\boldsymbol{u}$ at other position in the domain $\Omega$. In particular, when $\boldsymbol{u}(x)$ is smaller or larger than the values at other positions, it will increase or decrease, respectively. This is analogous to a diffusion process in which particles or substances move from high-concentration to low-concentration regions. It has been proved that a diffusion process converges to a saturating state in which the concentrations at all positions are the same. This suggests that $\boldsymbol{u}(x)$ tends to suffer from the over-smoothing issue.*

## 2.2 Random Walk Analysis of Over-smoothing

The diffusion process and random walk are closely related concepts, as diffusion can be seen as a collective behavior of numerous random walks performed by individual particles or molecules. Inspired by the analogy between the dynamics of $\boldsymbol{u}$ in Eqn 9 and a diffusion process, as well as the relationship between diffusion process and random walk, in this section, we show the connection

between the evolution of $\boldsymbol{u}$ and a random walk. By adopting a random walk perspective on graph neural network [58], we demonstrate that $\boldsymbol{u}(x)$ under the dynamics given in Eqn 9 suffers from over-smoothing.

Recall from the gradient flow in Eqn 9, by using Euler method discretization, after $k$ update steps starting from the initial $\boldsymbol{u}(x,0) = \boldsymbol{v}(x)$, with adaptive stepsize $\Delta t = 1/\int_\Omega \big(k(x,y) + k(y,x)\big)dy$, we obtain the following

$$\boldsymbol{u}(x, k\Delta t(x)) = \int_\Omega \frac{K(x,y)\boldsymbol{u}(y,(k-1)\Delta t(x))}{\int_\Omega K(x,y')dy'}dy. \tag{15}$$

Discretizing $\boldsymbol{u}(x, k\Delta t(x))$ and using Monte-Carlo approximation for the integrals in 15 , we attain

$$\boldsymbol{u}^{(k)}(i) = \sum_{j=1}^N \mathbf{A}_{ij}\boldsymbol{u}^{(k-1)}(j) \tag{16}$$

where $\mathbf{A}_{ij}$ is computed using the keys and queries as either $\mathrm{softmax}\big(\boldsymbol{k}(i)^\top \boldsymbol{k}(j)/\sqrt{D_{qk}}\big)$ or $\mathrm{softmax}\big(\boldsymbol{q}(i)^\top \boldsymbol{k}(j)/\sqrt{D_{qk}}\big)$. Let $\{\mathbf{B}^{(k)}(i)\}_{k\in K}$ be a random walk on $\{\boldsymbol{v}(i)\}_{i=1}^N$ as defined:

$$\mathbf{B}^{(0)}(i) = \boldsymbol{v}(i)$$
$$\mathbb{P}(\mathbf{B}^{(k+1)}(l) = \boldsymbol{v}(j)|\mathbf{B}^{(k)}(l) = \boldsymbol{v}(i)) = \mathbf{A}_{ij} \tag{17}$$

where $\mathbf{B}^{(k)}(n)$ is the random value of a $k$-step walk, starts at node $n$, and $\boldsymbol{v}(n)$ is the initial value at node $n$, respectively, for $n = 1, 2, \ldots, N$. The transition probability $\mathbf{A}$ is defined as above. To investigate the connection between the update process of $\boldsymbol{u}$ and the random walk defined in 17, we show that, for $i = 1, 2, \ldots, N$, after $k$ update steps as in 16, with initial value $\boldsymbol{u}^{(0)}(i) = \boldsymbol{v}(i)$, $\boldsymbol{u}(i)^{(k)}$ equals to the expected value of the $k$-step walk, starting at node $i$:

**Lemma 1.** *Let $\boldsymbol{u}^{(k)}(i)$ defined in 16 and $\{\mathbf{B}^{(k)}(i)\}_{k\in K}$ is the random walk defined by 17. Then*

$$\boldsymbol{u}^{(k)}(i) = \mathbb{E}[\mathbf{B}^{(k)}(i)]. \tag{18}$$

We next present the Lemma 2 which is necessary to show the convergence of $\boldsymbol{u}^{(k)}(i)$.

**Lemma 2.** *The random walk $\mathbf{B}^{(k)}(i)$ in 17 with the transition matrix $\mathbf{A}$ either be $\mathbf{A}_{ij} = \mathrm{softmax}\big(\boldsymbol{k}(i)^\top \boldsymbol{k}(j)/\sqrt{D_{qk}}\big)$ or $\mathbf{A}_{ij} = \mathrm{softmax}\big(\boldsymbol{q}(i)^\top \boldsymbol{k}(j)/\sqrt{D_{qk}}\big)$, has a unique stationary distribution $\boldsymbol{\pi} = [\pi_1, \pi_2, \ldots, \pi_N]$ such that $\pi_i := P(\mathbf{B}^{(k)}(j) = \boldsymbol{v}(i))$, for $i, j = 1, 2, \ldots, N$, $\sum_{i=1}^N \pi_i = 1$, and $\boldsymbol{\pi}^T = \boldsymbol{\pi}^T\mathbf{A}$.*

*If $\mathbf{A}_{ij} = \mathrm{softmax}\big(\boldsymbol{k}(i)^\top \boldsymbol{k}(j)/\sqrt{D_{qk}}\big)$, the stationary distribution is:*

$$\boldsymbol{\pi} = \left(\frac{d_1}{\sum_{j=1}^N d_j}, \frac{d_2}{\sum_{j=1}^N d_j}, \ldots, \frac{d_n}{\sum_{j=1}^N d_j}\right), \tag{19}$$

*where $d_i = \sum_{j=1}^N \exp\big(\boldsymbol{k}(i)^\top \boldsymbol{k}(j)/\sqrt{D_{qk}}\big)$, $\boldsymbol{k}(1), \boldsymbol{k}(2), \ldots, \boldsymbol{k}(N)$ are the key vectos.*

*In general, $\pi_i$ can be found by finding the left eigenvector of $\mathbf{A}$ corresponding to the dominant eigenvalue 1.*

From the Lemma 1 and Lemma 2, we see that, for all $i = 1, 2, \ldots, N$,

$$\boldsymbol{u}^{(k)}(i) = \mathbb{E}[\mathbf{B}^{(k)}(i)] = \sum_{j=1}^N \boldsymbol{v}(j)\mathbb{P}(\mathbf{B}^{(k-1)}(i) = \boldsymbol{v}(j)) \to \sum_{j=1}^N \pi_j \boldsymbol{v}(j) =: \bar{\boldsymbol{v}}. \tag{20}$$

as $k \to \infty$. This shows that when $k$ increases, $\boldsymbol{u}(i)^{(k)}$ converges to a constant vector, indicating that $\boldsymbol{u}(x)$, under the dynamic in 9, suffers from over-smoothing.

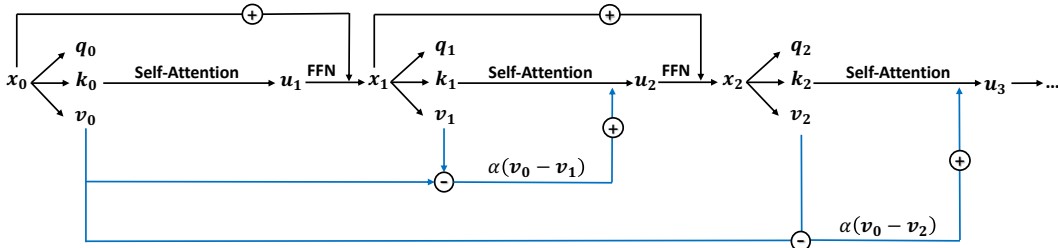

Figure 2: Our proposed NeuTRENO model adds a proportion of the difference between the values of the first and that of the current layer to the self-attention's output at each layer.

# 3 NeuTRENO: Mitigating the Over-smoothing in Transformers via Minimizing a Regularized Functional

In Section 2.1, we have shown that self-attention implicitly performs a gradient descent step to minimize the nonlocal functional $J(\boldsymbol{u})$ in Eqn. 5, which results in the diffusive characteristics of $\boldsymbol{u}$ and causes the over-smoothing phenomenon in transformers, as proved in Section 2.2. Fortunately, our objective is not to minimize $J(\boldsymbol{u})$ but the energy/regularized functional $E(\boldsymbol{u}, \boldsymbol{f})$ defined by Eqn. 5. This regularized functional consists of not only $J(\boldsymbol{u})$ but also the convex fidelity term $G(\boldsymbol{u}, \boldsymbol{f}) = \frac{\lambda}{2} \int_{\Omega} \|\boldsymbol{u}(x) - \boldsymbol{f}(x)\|_2^2 dx$. This fidelity term aims to preserve the relevant information in the observed noisy signal $\boldsymbol{f}(x)$ by penalizing solution $\boldsymbol{u}(x)$ that deviates significantly from $\boldsymbol{f}(x)$, thereby mitigating the effects of over-smoothing caused by minimizing $J(\boldsymbol{u})$.

In this section, we will derive our Neural Transformer with a Regularized Nonlocal Functional (NeuTRENO) by minimizing the regularized functional $E(\boldsymbol{u}, \boldsymbol{f})$. We then provide a theoretical result to prove that NeuTRENO does not suffer from over-smoothing. Recall from Eqn. 5 that $E(\boldsymbol{u}, \boldsymbol{f})$ is given by

$$E(\boldsymbol{u}, \boldsymbol{f}) = J(\boldsymbol{u}) + G(\boldsymbol{u}, \boldsymbol{f}) = J(\boldsymbol{u}) + \frac{\lambda}{2} \int_{\Omega} \sum_{j=1}^{D} (u_j(x) - f_j(x))^2 dx$$

Following a similar derivation as in Section 2.1 (see Appendix C for the detailed derivation), we obtain the following gradient flow when minimizing $E(\boldsymbol{u}, \boldsymbol{f})$ using gradient descent

$$\frac{d\boldsymbol{u}(x,t)}{dt} = -\nabla_{\boldsymbol{u}} E(\boldsymbol{u}, \boldsymbol{f}) = -\nabla_{\boldsymbol{u}} J(\boldsymbol{u}) - \lambda\big(\boldsymbol{u}(x) - \boldsymbol{f}(x)\big), \tag{21}$$

**NeuTRENO-attention is an Euler Discretization of the Gradient Flow Given in 21.** Following the similar derivation in Section 2.1, we discretize the gradient flow in Eqn. 21 using the Euler method [21] with step size $\Delta t(x) = 1/\int_{\Omega}\big(k(x,y) + k(y,x)\big)dy$ and initializing $\boldsymbol{u}$ at $t = 0$ with $\boldsymbol{v}(x)$, i.e., $\boldsymbol{u}(x,0) = \boldsymbol{v}(x)$. Choosing $\lambda = \tilde{\lambda}/\Delta t(x)$, we obtain the following update

$$\boldsymbol{u}(x, \Delta t(x)) = \boldsymbol{u}(x,0) - \Delta t(x)\nabla_{\boldsymbol{u}} J - \lambda\Delta t(x)\big(\boldsymbol{u}(x,0) - \boldsymbol{f}(x)\big)$$
$$= \int_{\Omega} \frac{K(x,y)\boldsymbol{v}(y)}{\int_{\Omega} K(x,y')dy'} dy + \tilde{\lambda}\big(\boldsymbol{f}(x) - \boldsymbol{v}(x)\big). \tag{22}$$

We choose the observed noisy signal $\boldsymbol{f}(x) = \boldsymbol{v}^0(x)$ where $\boldsymbol{v}^0(x)$ is $\boldsymbol{v}(x)$ at the first layer in the transformer model. The update in Eqn. 22 becomes

$$\boldsymbol{u}(x, \Delta t(x)) = \int_{\Omega} \frac{K(x,y)\boldsymbol{v}(y)}{\int_{\Omega} K(x,y')dy'} dy + \tilde{\lambda}\big(\boldsymbol{v}^0(x) - \boldsymbol{v}(x)\big). \tag{23}$$

Applying the Monte-Carlo method to approximate the integrals in Eqn. 23 and discretizing $\boldsymbol{u}(x, \Delta t(x))$, $\boldsymbol{v}(x)$, and $\boldsymbol{v}^0(x)$ on a 1-D grid, we attain the following new formula for calculating symmetric self-attention:

$$\boldsymbol{u}(i) = \sum_{j=1}^{N} \mathrm{softmax}\Big(\boldsymbol{k}(i)^{\top}\boldsymbol{k}(j)/\sqrt{D_{qk}}\Big)\boldsymbol{v}(j) + \tilde{\lambda}(\boldsymbol{v}^0(i) - \boldsymbol{v}(i)), \quad i = 1, \ldots, N. \tag{24}$$

Table 1: Top-1 and Top-5 accuracy (%) of NeuTRENO DeiT vs. DeiT on the ImageNet benchmark. We also present the performance of adapting NeuTRENO to the pre-trained DeiT baseline, NeuTRENO Adaptation. In addition, we compare NeuTRENO with FeatScale [65] and incorporate our method with FeatScale model.

| Model/Metric | Top-1 Acc (%) | Top-5 Acc (%) |
|---|---|---|
| *Softmax DeiT* | 72.17 | 91.02 |
| NeuTRENO-DeiT | **73.01** | **91.56** |
| NeuTRENO Adaptation | 72.63 | 91.38 |
| *DeiT + FeatScale* | 72.346 | 91.22 |
| NeuTRENO DeiT + FeatScale | **73.23** | **91.73** |

Its corresponding asymmetric self-attention is obtained by replacing the key vectors $\boldsymbol{k}(i)$ with the query vectors $\boldsymbol{q}(i)$, $i = 1, \ldots, N$, and given by

$$\boldsymbol{u}(i) = \sum_{j=1}^{N} \mathrm{softmax}\Big(\boldsymbol{q}(i)^{\top} \boldsymbol{k}(j)/\sqrt{D_{qk}}\Big)\boldsymbol{v}(j) + \tilde{\lambda}(\boldsymbol{v}^0(i) - \boldsymbol{v}(i)), \ \ i = 1, \ldots, N. \quad (25)$$

Leveraging Eqn. 25, we define the Neural Transformer with a Regularized Nonlocal Functional (NeuTRENO) as follows.

**Definition 1** (Neural Transformer with a Regularized Nonlocal Functional (NeuTRENO)). *Given a set of key and value vectors* $\{\boldsymbol{k}^{\ell}(j), \boldsymbol{v}^{\ell}(j)\}_{j=1}^{N}$ *in each layer* $\ell$, $\ell = 1, \ldots, L$, *for each query vector* $\boldsymbol{q}^{\ell}(i)$, $i = 1, \ldots, N$, *in the same layer, the self-attention unit at layer* $\ell$ *in a Neural Transformer with a Regularized Nonlocal Functional (NeuTRENO) computes the corresponding output vector* $\boldsymbol{u}^{\ell}(i)$ *of the query* $\boldsymbol{q}^{\ell}(i)$ *by the following attention formula:*

$$\boldsymbol{u}^{\ell}(i) = \sum_{j=1}^{N} \mathrm{softmax}\Big(\boldsymbol{q}^{\ell}(i)^{\top} \boldsymbol{k}^{\ell}(j)/\sqrt{D_{qk}}\Big)\boldsymbol{v}^{\ell}(j) + \tilde{\lambda}(\boldsymbol{v}^0(i) - \boldsymbol{v}^{\ell}(i)), \ \ i = 1, \ldots, N. \quad (26)$$

*where* $\boldsymbol{v}^0(1), \ldots \boldsymbol{v}^0(N) \in \mathbb{R}^D$ *are the value vectors in the first layer of NeuTRENO.*

Fig. 2 illustrates the architecture of NeuTRENO.

**Proposition 1.** *The evolution of* $\boldsymbol{u}(x)$ *under the dynamic in 21 does not converge to a constant vector.*

Proposition 1 indicates that our NeuTRENO mitigates the over-smoothing issue, suggesting the benefit of our method. The proof for Proposition 1 is given in Appendix B.3.

## 4 Experimental Results

In this section, we empirically demonstrate the advantages of our proposed NeuTRENO approach across various tasks, including ImageNet classification [15], ADE20K image segmentation [73], and language modeling on the WikiText-103 [42]. Our aim to show: (i) NeuTRENO significantly outperforms the transformer baseline with softmax-attention defined in 2 across various tasks; moreover, NeuTRENO surpass FeatScale, a vision transformer that addresses over-smoothing, combining NeuTRENO with FeatScale is beneficial; (ii) the advantages of incorporating our proposed method with pre-trained models. We also demonstrate the benefits of our NeuTRENO in the symmetry setting and we point to Appendix D for the results. Throughout our experiments, we compare the performance of our proposed models with baselines of the same configuration. For additional details regarding datasets, models, and training procedures, please refer to Appendix A.

**Object classification on ImageNet.** To demonstrate the advantage of our NeuTRENO method, we compare it with the DeiT baseline [59] on the ImageNet image classification task. Our NeuTRENO DeiT surpasses the DeiT baseline, as shown in Table 1. Notably, our NeuTRENO DeiT achieves significantly higher performance in terms of both Top-1 Accuracy and Top-5 Accuracy. We also compare our method with FeatScale [65], a vision transformer model addressing over-smoothing (see Table 1). Our NeuTRENO significantly outperforms FeatScale, and combining NeuTRENO with FeatScale leads to substantial improvements. These results confirm the benefits of our model.

**Image Segmentation on ADE20K dataset.** To further validate the advantages of our proposed methods, we compare the performance of the Segmenter models [56] using the NeuTRENO DeiT

Table 2: Single-scale (SS) MIoU and multi-scale MIoU (MS) of the NeuTRENO DeiT vs. the DeiT on the ADE20K image segmentation.

| Model/Metric | SS MIoU | MS MIoU (%) |
|---|---|---|
| *Softmax DeiT* | 35.72 | 36.68 |
| NeuTRENO DeiT | **37.24** | **38.06** |

Table 3: Test and valid perplexity (Test PPL and Valid PPL) on WikiText-103 of NeuTRENO compared to the softmax transformer. Our proposed method achieves a significantly better performance PPL than the baseline.

| Method/Metric | Valid PPL | Test PPL |
|---|---|---|
| *Softmax Transformer* | 33.15 | 34.29 |
| NeuTRENO | **32.60** | **33.70** |

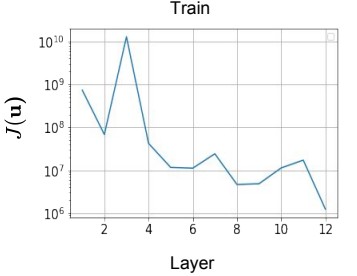 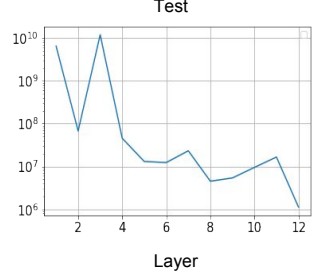

Figure 3: The average value of functional $J(\boldsymbol{u})$ over 1000 training (Left) samples and test (Right) samples. When softmax attention is applied, the functional decreases as the depth of the trained DeiT increases.

and DeiT backbones the on ADE20K image segmentation task [72], as shown in Table 2. The results demonstrate the substantial performance improvements achieved by utilizing the NeuTRENO DeiT backbone over the DeiT backbone, in terms of both single-scale (SS) MIoU and multi-scale (MS) MIoU metrics. These results strongly emphasize the effectiveness of our NeuTRENO approach in enhancing image segmentation performance.

**Language Model on WikiText-103.** In addition to computer vision tasks, we also evaluate the effectiveness of our model on a large-scale natural language processing application, specifically language modeling on WikiText-103. Our NeuTRENO language model demonstrates better performance in terms of both test perplexity and valid perplexity when compared to the softmax transformer language model [68]. These findings, combined with the results obtained across various tasks, empirically confirm the significant benefits of our NeuTRENO models.

**Combine with pre-trained models.** Furthermore, our proposed method is also beneficial to combine with pre-trained models. To empirically demonstrate that we incorporate NeuTRENO with pre-trained DeiT and fine-tune on the ImageNet dataset with one-third number of epochs that are used in training. The result is presented in Table 1, showing that combined with our method improves both the Top-1 and Top-5 accuracies of the pre-trained models.

## 5 Empirical Analysis

**Applying Softmax-Attention Reduces the functional $J(\boldsymbol{u})$.** We present evidence supporting that the employment of softmax attention minimizes the functional $J(\boldsymbol{u})$. Initially, we observe that the average cosine similarity between the numerical approximation of $\nabla_{\boldsymbol{u}} J(\boldsymbol{u})$ using symmetric or asymmetric kernel $K(x, y)$ for both the trained Sym-DeiT (using symmetric self-attention 14) and DeiT models, closed 1, as shown in Table 4. This suggests that reversing the direction of the asymmetric approximation effectively decreases $J(\boldsymbol{u})$. Considering that softmax attention takes steps in this reversed direction numerically, its application leads to a reduction in $J(\boldsymbol{u})$. This is further substantiated by Fig. 3, which demonstrates a decrease in $J(\boldsymbol{u})$ as the depth of the trained DeiT increases when softmax attention is employed. More details of this analysis are in Appendix E

**Over-smoothing Analysis.** We empirically illustrate the effectiveness of NeuTRENOs in mitigating the over-smoothing problem in transformers. Fig. 1 compares the cosine similarity between token representations across layers for both NeuTRENO and softmax baseline models, specifically focusing on the Imagenet classification task (Left) and ADE20K image segmentation (Right). The token

Table 4: The average cosine similarity between the numerical approximation of $\nabla J(\boldsymbol{u})(x)$ using symmetric or asymmetric kernel $K(x,y)$, for the trained Sym-DeiT and softmax DeiT models. The metric is evaluated on 1000 training and 1000 test data samples. The average score close to 1 shows a strong alignment between symmetric and asymmetric gradient approximations, suggesting that reversing the direction of the asymmetric approximation effectively reduces the functional $J(\boldsymbol{u})$.

| Model | Training data | Test data |
|---|---|---|
| Sym-DeiT | 0.982 | 0.976 |
| Softmax DeiT | 0.973 | 0.964 |

features extracted by NeuTRENOs exhibit significantly lower similarity, particularly in the final layers. This finding highlights the ability of NeuTRENOs to address the over-smoothing issue and improve the diversity of token representations. We provide more details of this analysis in Appendix E.

## 6 Related Work

**Over-smoothing in Transformers.** Over-smoothing in deep transformers has been observed in various domains and applications from natural language processing [55] to computer vision [65, 18]. In vision tasks, [74] observes that the performance of the vision transformer (ViT [20]) quickly saturates as more layers are added to the model. Moreover, experiments in [74] show that the 32-layer ViT underperforms the 24-layer ViT, indicating the difficulty of ViTs in gaining benefits from deeper architectures. The authors point out that over-smoothing results in this phenomenon by causing the token representations to become identical when the model grows deeper. Based on this observation, the authors propose a cross-head communication method that helps enhance the diversity of both token representations and attention matrices. Furthermore, it has been shown in [60] that the training of ViT models encounters instability with greater depths. [25] proposes that this instability arises from the over-smoothing, where token representation for patches within an image becomes progressively alike as the model's depth increases. To explain this issue, [65] finds out that self-attention acts as a low-pass filter and smoothens the token representations in ViTs. This leads to the proposal of the FeatScale method [65], which regulates feature frequencies, whether low or high, to counteract the consequences of over-smoothing.

In addition, [55] observes the phenomenon in BERT [16], a deep language model, and explores over-smoothing through the graph perspective. The work utilizes hierarchical fusion strategies by preserving the output of self-attention through all layers, which is memory-costly. On the other hand, [65, 18] investigate over-smoothing in the image domain through the lens of Fourier spectrum, showing that self-attentions are low-pass filters, retaining only low-frequency, causing over-smoothed outputs. Our work is an orthogonal explanation of the previous work. We focus on developing a variational denoising framework to understand the self-attention of transformers as a gradient descent approximation of a functional. Our new finding explains the over-smoothing issue of transformers due to self-attention minimizing a functional and inspires us to derive the novel NeuTRENO method to overcome over-smoothing.

**Nonlocal Functionals for Image Processing.** Total variation [51] is well-known as an image-denoising technique. It denoises a noisy image by solving a constraint optimization problem. The method is also related to PDE-flow-based image-denoising techniques [24], namely isotropic and anisotropic diffusion [67] models. The method is edge preserving, meaning to avoid over-blurring edges' information [7]. Nonlocal functionals [35, 24] is considered as an extension of total variation to a nonlocal scale. Nonlocal functional and edge preservation properties are the motivation of our work to explain and overcome over-smoothing in transformers.

## 7 Concluding Remarks

In this paper, we establish a nonlocal variational denoising framework for self-attention. From this variational perspective, we explain over-smoothing in self-attention, which hinders the representation capacity of transformer models. We also derive the novel Neural Transformer with a Regularized Nonlocal Functional (NeuTRENO) to alleviate the over-smoothing. We empirically verify the benefits of NeuTRENO with a wide range of large-scale applications including ImageNet object classification, ADE20K object segmentation, and WikiText-103 language modeling. A limitation of our paper is that the privacy-preserving of NeuTRENO has not been addressed. It is interesting to explore if regularized nonlocal functional can also help improve the privacy-preserving of transformer models. We leave this exciting research idea as future work.

## Acknowledgments and Disclosure of Funding

RGB acknowledges support from the NSF grants CCF-1911094, IIS-1838177, and IIS-1730574;ONR grants N00014-18-12571, N00014-20-1-2534, and MURI N00014-20-1-2787; AFOSR grant FA9550-22-1-0060; and a Vannevar Bush Faculty Fellowship, ONR grant N00014-18-1-2047. TMN acknowledges support from his start-up grant at the National University of Singapore (Grant Number: A-0009807-00-00).

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

# Supplement to "Mitigating Over-smoothing in Transformers via Regularized Nonlocal Functionals"

**Table of Contents**

## A   Additional Details on the Experiments in Section 4

This section provides datasets, models, and training details for experiments in Section 4. The code to reproduce our experimental results is included in our Supplementary Material submission.

### A.1 Image Classification on Imagenet

**Datasets and Metrics.** The ImageNet dataset [15, 52] comprises $1.28$ million training images and $50,000$ validation images, encompassing the classification of 1000 categories. The evaluation metrics used for performance assessment are the top-1 and top-5 accuracies.

**Models and Baselines.** Our baseline model is the DeiT-tiny model [59], which consists of 12 transformer layers, 3 attention heads per layer, and a model dimension of 192. For model setting and setting and configuration, we follow [59]. Their implementation is available at https://github.com/facebookresearch/deit. The $\tilde{\lambda}$ used for our NeuTRENO method is $0.6$.

### A.2 Image Segmentation on ADK20 dataset

**Datasets and Metrics.** The ADE20K dataset is recognized for its inclusion of challenging scenes with fine-grained labels, making it one of the most demanding semantic segmentation datasets. The training set consists of 20,210 images encompassing 150 semantic classes. Additionally, there are 2,000 images in the validation set and 3,352 images in the test set. This in task the Single-scale mean Intersection over Union (SS mIoU) and the Multi-scale (MS mIoU).

**Models and baselines.** The training configuration and setting for our models are followed by [56]. The baseline model is finetuned with the pretrained DeiT-tiny backbone while our segmenter model used the pretrained NeuTRENO DeiT-tiny, with $\tilde{\lambda} = 0.6$.

### A.3 Language Modeling on WikiText-103

**Datasets and Metrics.** The WikiText-103 dataset consists of articles extracted from Wikipedia and is specifically designed to capture long contextual dependencies. The training set comprises approximately $28,000$ articles, totaling $103$ million running words. Each article contains text blocks consisting of approximately $3,600$ words. The validation and test sets contain $218,000$ and $246,000$ running words, respectively, with each set consisting of 60 articles and approximately $268,000$ words. Our experiment follows the standard setting [42, 53], which involves dividing the training data into independent long segments of $L$ words. For evaluation, we employ a batch size of 1 and process the text sequence using a sliding window of size $L$. When computing perplexity (PPL), we consider only the last position, except for the first segment where all positions are evaluated, following the approach in [2, 53].

**Models and baselines.** For our language modeling implementation, we rely on the publicly available code https://github.com/IDSIA/lmtool-fwp developed by [53]. In our experiments, we set the dimensions of keys, values, and queries to 128, while the training and evaluation context length is set to 256. In this experiment, $\tilde{\lambda} = 0.4$ yields the best performance of NeuTRENO language model.

## B Technical Proofs

### B.1 Proof of Lemma 1

For all $i = 1, \ldots, N$, we have $\mathbb{E}[\mathbf{B}^{(0)}(i)] = \boldsymbol{v}(i)$. Assume that $\mathbb{E}[\mathbf{B}^{(k)}(i)] = \boldsymbol{u}^{(k)}(i)$, then

$$
\begin{aligned}
\mathbb{E}[\mathbf{B}^{(k+1)}(i)] &= \sum_{j=1}^{N} \boldsymbol{v}(j) \mathbb{P}(\mathbf{B}^{(k+1)}(i) = \boldsymbol{v}(j)) \\
&= \sum_{j=1}^{N} \boldsymbol{v}(j) \sum_{l=1}^{N} \mathbb{P}(\mathbf{B}^{(k+1)}(i) = \boldsymbol{v}(j)|\mathbf{B}^{(1)}(i) = \boldsymbol{v}(l)) \mathbb{P}(\mathbf{B}^{(1)}(i) = \boldsymbol{v}(l)) \\
&= \sum_{j=1}^{N} \boldsymbol{v}_j \sum_{l=1}^{N} \mathbb{P}(\mathbf{B}^{(k)}(l) = \boldsymbol{v}(j)) \mathbb{P}(\mathbf{B}^{(1)}(i) = \boldsymbol{v}(l)|\mathbf{B}^{(0)}(i) = \boldsymbol{v}(i)) \\
&= \sum_{j=1}^{N} \boldsymbol{v}(j) \sum_{l=1}^{N} \mathbf{A}_{il} \mathbb{P}(\mathbf{B}^{(k)}(l) = \boldsymbol{v}(j)) \\
&= \sum_{l=1}^{N} \mathbf{A}_{il} \mathbb{E}[\mathbf{B}^{(k)}(l)] = \sum_{l=1}^{N} \mathbf{A}_{il} \boldsymbol{u}^{(k)}(l) \\
&= \boldsymbol{u}^{(k+1)}(i).
\end{aligned}
$$

Thus, by induction, we obtain the conclusion of the lemma.

## B.2 Proof of Lemma 2

Since the transition matrix $\mathbf{A} \in \mathbb{R}^{N \times N}$ is right-stochastic, its largest eigenvalue is 1 (see Theorem 4.1 in [5]). Also, $\mathbf{A}$ is a regular positive matrix since its elements are positive. Thus, the Perron-Frebenius theorem [8] implies the existence of a unique probability distribution $\boldsymbol{\pi}$, which is a positive left eigenvector of the transition matrix $\mathbf{A}$ associated with its largest eigenvalue 1. In particular, in the case of symmetricity constraint, $\boldsymbol{\pi}$ can be chosen as follows

$$\boldsymbol{\pi} = \left( \frac{d_1}{\sum_{j=1}^N d_j}, \frac{d_2}{\sum_{j=1}^N d_j}, \cdots, \frac{d_n}{\sum_{j=1}^N d_j} \right),$$

where $d_i = \sum_{j=1}^N \exp\left( \boldsymbol{k}(i)^\top \boldsymbol{k}(j) / \sqrt{D_{qk}} \right)$. It is easy to see that

$$\sum_{i=1}^N \pi_i \mathbf{A}_{ij} = \sum_{i=1}^N \frac{d_i}{\sum_{l=1}^N d_l} \frac{\exp\left( \boldsymbol{k}(i)^\top \boldsymbol{k}(j) / \sqrt{D_{qk}} \right)}{d_i}$$

$$= \frac{\sum_{i=1}^N \left( \exp\left( \boldsymbol{k}(i)^\top \boldsymbol{k}(j) / \sqrt{D_{qk}} \right) \right)}{\sum_{l=1}^N d_l}$$

$$= \frac{d_j}{\sum_{l=1}^N d_l} = \pi_j.$$

As a consequence, $\boldsymbol{\pi}$ must be the unique stationary distribution of the random walk $\{\mathbf{B}^{(k)}(i)\}_{k \in K}$. This concludes the proof.

## B.3 Proof of Proposition 1

Recall from the gradient flow in Eqn 21, by using the method of Euler discretization, after $k$ update steps starting from the initial $\boldsymbol{u}(x, 0) = \boldsymbol{v}(x)$ with adaptive stepsize $\Delta t = 1/\int_\Omega \left( k(x, y) + k(y, x) \right) dy$ and by choosing $\lambda = \tilde{\lambda}/\Delta t(x)$, we obtain the following

$$\boldsymbol{u}(x, k\Delta t(x)) = \boldsymbol{u}(x, (k-1)\Delta t(x)) - \Delta t(x) \nabla_{\boldsymbol{u}} J - \lambda \Delta t(x) \left( \boldsymbol{u}(x, (k-1)\Delta t(x)) - \boldsymbol{f}(x) \right)$$

$$= \int_\Omega \frac{K(x, y) \boldsymbol{u}(y, (k-1)\Delta t(x))}{\int_\Omega K(x, y') dy'} dy + \tilde{\lambda} \left( \boldsymbol{f}(x) - \boldsymbol{u}(x, (k-1)\Delta t(x)) \right). \qquad (27)$$

Discretizing $\boldsymbol{u}(x, k\Delta t(x))$ and using Monte-Carlo approximation for the integrals in 27 , we obtain

$$\boldsymbol{u}^{(k)}(i) = \sum_{j=1}^N \mathbf{A}_{ij} \boldsymbol{u}^{(k-1)}(j) + \tilde{\lambda} \left( \boldsymbol{f}(i) - \boldsymbol{u}^{(k-1)}(i) \right), \qquad (28)$$

where $\mathbf{A}_{ij}$ is computed using the keys and queries as either $\mathrm{softmax}\left( \boldsymbol{k}(i)^\top \boldsymbol{k}(j) / \sqrt{D_{qk}} \right)$ or $\mathrm{softmax}\left( \boldsymbol{q}(i)^\top \boldsymbol{k}(j) / \sqrt{D_{qk}} \right)$.

Suppose that $\boldsymbol{u}^{(k)}(i)$, defined as Eqn. 28, converges to a constant vector $\bar{\boldsymbol{u}}$ as $k \to \infty$. We have

$$\boldsymbol{u}^{(k+1)}(i) - \boldsymbol{u}^{(k+1)}(j)$$

$$= \sum_{l=1}^{N} \mathbf{A}_{il}\boldsymbol{u}^{(k)}(l) - \sum_{l=1}^{N} \mathbf{A}_{jl}\boldsymbol{u}^{(k)}(l) + \tilde{\lambda}(\boldsymbol{u}^{(k)}(j) - \boldsymbol{u}^{(k)}(i)) + \tilde{\lambda}(\boldsymbol{f}(i) - \boldsymbol{f}(j))$$

$$= \Big(\sum_{l=1}^{N} \mathbf{A}_{il}\boldsymbol{u}^{(k)}(l) - \boldsymbol{u}^{(k)}(i)\sum_{l=1}^{N} \mathbf{A}_{il}\Big) - \Big(\sum_{l=1}^{N} \mathbf{A}_{jl}\boldsymbol{u}^{(k)}(l) - \boldsymbol{u}^{(k)}(j)\sum_{l=1}^{N} \mathbf{A}_{jl}\Big) \tag{29}$$

$$+ (\tilde{\lambda} - 1)(\boldsymbol{u}^{(k)}(j) - \boldsymbol{u}^{(k)}(i)) + \tilde{\lambda}(\boldsymbol{f}(i) - \boldsymbol{f}(j))$$

$$= \sum_{l=1}^{N} \mathbf{A}_{il}(\boldsymbol{u}^{(k)}(l) - \boldsymbol{u}^{(k)}(i)) - \sum_{l=1}^{N} \mathbf{A}_{jl}(\boldsymbol{u}^{(k)}(l) - \boldsymbol{u}^{(k)}(j)) + (\tilde{\lambda} - 1)(\boldsymbol{u}^{(k)}(j) - \boldsymbol{u}^{(k)}(i))$$

$$+ \tilde{\lambda}(\boldsymbol{f}(i) - \boldsymbol{f}(j))$$

Since $\boldsymbol{u}^{(k)}(i) \to \bar{\boldsymbol{u}}$, for $i = 1, 2, \ldots, N$, as $k \to \infty$, we have $\begin{cases} (\boldsymbol{u}^{(k+1)}(i) - \boldsymbol{u}^{(k+1)}(j)) \to \mathbf{0} \\ (\boldsymbol{u}^{(k)}(l) - \boldsymbol{u}^{(k)}(i)) \to \mathbf{0} \\ (\boldsymbol{u}^{(k)}(l) - \boldsymbol{u}^{(k)}(j)) \to \mathbf{0} \\ (\boldsymbol{u}^{(k)}(j) - \boldsymbol{u}^{(k)}(i)) \to \mathbf{0} \end{cases}$

as $k \to \infty$. This is a contradiction since while the LHS of 29 approaches $\mathbf{0}$, its RHS approaches $\tilde{\lambda}(\boldsymbol{f}(i) - \boldsymbol{f}(j))$, which is not $\mathbf{0}$ in general. Thus, we obtain the conclusion of Proposition 1.

## C   Derivation of Gradient of E as Given in Eqn. 21

Taking the gradient of $E(\boldsymbol{u}, \boldsymbol{f})$ with respect to $\boldsymbol{u}$, we obtain

$$\nabla_{\boldsymbol{u}} E = \nabla_{\boldsymbol{u}} J + \left[\frac{\partial G}{\partial u_1}, \frac{\partial G}{\partial u_2}, \ldots, \frac{\partial G}{\partial u_D}\right]^T. \tag{30}$$

The partial derivative $\partial G/\partial u_j$, $j = 1, 2, \ldots, D$, is defined through its dot product with an arbitrary function $h_j \in L^2(\Omega)$ as follows

$$\frac{\partial G}{\partial u_j} \cdot h_j(x) = \frac{d}{d\tau} G(u_j + \tau h_j)\big|_{\tau=0}$$

$$= \frac{\lambda}{2}\left(\frac{d}{d\tau}\int_{\Omega}(u_j(x) - f_j(x) + \tau h_j(x))^2 dx\right)\bigg|_{\tau=0}$$

$$= \lambda \int_{\Omega}(u_j(x) - f_j(x))h_j(x)dx.$$

Thus, the Frechet derivative of F with respect to $u_j$ is given by

$$\frac{\partial G}{\partial u_j} = \lambda(u_j(x) - f_j(x)) \tag{31}$$

Substituting the formula for $\partial G/\partial u_j$ in Eqn. 31 into Eqn. 30 for $\nabla_{\boldsymbol{u}} E(\boldsymbol{u}, \boldsymbol{f})$, we obtain the following gradient flow

$$\frac{d\boldsymbol{u}(x,t)}{dt} = -\nabla_{\boldsymbol{v}} E(\boldsymbol{u}, \boldsymbol{f}) = -\nabla_{\boldsymbol{u}} J(\boldsymbol{u})(x) + \lambda\big(\boldsymbol{f}(x) - \boldsymbol{u}(x)\big), \tag{32}$$

where $t$ is a dummy time variable and $-\nabla_{\boldsymbol{u}} J(\boldsymbol{u})$ is defined as in 9.

## D   Results of Symmetric Setting

In this section, we show that NeuTRENO significantly improves the performance of a symmetric transformer baseline, which utilizes symmetric self-attention. We refer to the DeiT with symmetric attention, defined in 14, as Sym-DeiT and the Sym-DeiT combined with our NeuTRENO method as Sym-NeuTRENO DeiT.

**Object classification on Imagenet** To further illustrate the advantage of our NeuTRENO method, we compare Sym-NeuTRENO DeiT with the Sym-DeiT baseline on the ImageNet image

Table 5: Top-1 and Top-5 accuracy (%) of Sym-NeuTRENO DeiT vs. Sym-DeiT on the ImageNet classification task. The Sym-NeuTRENO DeiT models significantly outperform the Sym-DeiT in terms of accuracy, indicating the benefit of NeuTRENO method.

| Model/Metric | Top-1 Acc (%) | Top-5 Acc (%) |
|---|---|---|
| *Sym-DeiT* | 71.14 | 90.54 |
| Sym-NeuTRENO DeiT | **72.07** | **91.22** |

Table 6: Single-scale (SS) MIoU and multi-scale (MS) MIoU of the Sym-NeuTRENO DeiT vs. Sym-DeiT. The Sym-NeuTRENO DeiT model is beneficial since they significantly outperform the Sym-DeiT.

| Model/Metric | SS MIoU | MS MIoU (%) |
|---|---|---|
| *Sym-DeiT* | 35.18 | 36.00 |
| Sym-NeuTRENO DeiT | **35.68** | **36.39** |

classification task. Our Sym-NeuTRENO DeiT outperforms the Sym-DeiT baseline, as shown in Table 5. Notably, the Sym-NeuTRENO DeiT achieves higher performance in terms of both top-1 accuracy and top-5 accuracy than Sym-DeiT baseline. These results further confirm the benefits of our proposed NeuTRENO model.

**Image Segmentation on ADE20K dataset** We also compare the performance of the Segmenter models [56] using the Sym-NeuTRENO DeiT backbone with models using the Sym-DeiT backbone on ADE20K image segmentation [72], as shown in Table 6. The results demonstrate the substantial performance improvements achieved by utilizing the Sym-NeuTRENO DeiT backbone compared to the Sym-DeiT backbone in terms of both single-scale (SS) MIoU and multi-scale (MS) MIoU metrics. This result further validates the advantages of our NeuTRENO models in enhancing image segmentation performance in the symmetric setting.

# E    Additional Details on the Empirical Analysis in Section 5

In this section, we provide the details for the empirical analysis in Section 5.

## E.1    Average Cosine Similarity between Gradient Approximations

To produce the results in Table 4, we derive the approximation for the gradient $\nabla_{\boldsymbol{u}} J(\boldsymbol{u})$, from Eqn 9, at time $t = 0$:

$$\nabla_{\boldsymbol{u}} J(\boldsymbol{u}) = \int_{\Omega} \big(\boldsymbol{u}(x,0) - \boldsymbol{u}(y,0)\big) K(x,y) dy = \int_{\Omega} \big(\boldsymbol{v}(x) - \boldsymbol{v}(y)\big) K(x,y) dy,$$

where $K(x,y) := k(x,y) + k(y,x)$. Using Monte-Carlo approximation for the integral and choosing $K(x,y) = \exp\big(\boldsymbol{k}(x)^T \boldsymbol{k}(y)/\sqrt{D_{qk}}\big)$, the symmetric approximation of the gradient is derived as $\sum_{j=1}^{N} \big(\boldsymbol{v}(i) - \boldsymbol{v}(j)\big) \exp\big(\boldsymbol{k}(i)^T \boldsymbol{k}(j)/\sqrt{D_{qk}}\big)$. Otherwise, by choosing $K(x,y) = \exp\big(\boldsymbol{q}(x)^T \boldsymbol{k}(y)/\sqrt{D_{qk}}\big)$, the assymmetric approximation of the gradient is derived as $\sum_{j=1}^{N} \big(\boldsymbol{v}(i) - \boldsymbol{v}(j)\big) \exp\big(\boldsymbol{q}(i)^T \boldsymbol{k}(j)/\sqrt{D_{qk}}\big)$. In this analysis, we take the dot product between the symmetric and asymmetric approximation of the gradient $\nabla_{\boldsymbol{u}} J(\boldsymbol{u})$ and average these dot products over positions. We finally report the average cosine similarity over 1000 training data and 1000 test data, as shown in Table 4.

## E.2    Average Value of Function

In order to report the average value of function $J(\boldsymbol{u})$ in Fig. 3, we follow the process of computing $J(\boldsymbol{u})$ for 1000 data points for each transformer block. Subsequently, the average value is reported for each layer. This procedure is carried out for both the training and test datasets.

## E.3    Over-smoothing Analysis

The average cosine similarity between all pairs of token's representations $(\boldsymbol{x}_i, \boldsymbol{x}_j)$ in a sequence is computed as

$$\frac{1}{N(N-1)} \sum_{i \neq j} \frac{\boldsymbol{x}_i^T \boldsymbol{x}_j}{\|\boldsymbol{x}_i\|_2 \|\boldsymbol{x}_j\|_2}.$$

The result is then averaged over 1000 randomly chosen test data in ImageNet and ADE20K. The result is then reported for each layer, as in Fig. 1.

Table 7: Top-1 and Top-5 accuracy (%) of NeuTRENO DeiT-small vs. DeiT-small on the ImageNet benchmark. The NeuTRENO DeiT-small significantly outperform the DeiT-small in terms of accuracy. We also compare NeuTRENO DeiT-small with DeiT plus FeatScale, a vision transformer model that addresses over-smoothing, showing the advantage of NeuTRENO. The accuracies reported in [59] for DeiT-small and [65] for DeiT-small plus FeatScale, respectively, are in parentheses.

| Model/Metric | Top-1 Acc (%) | Top-5 Acc (%) |
|---|---|---|
| *DeiT-small* | 79.97 (79.9) | 95.05 (95.0) |
| DeiT-small + FeatScale | 79.96 (80.9) | 95.06 |
| NeuTRENO DeiT-small | **80.68** | **95.30** |

Table 8: Accuracy of NeuTRENO vs.Kernel Transformerr on the CIFAR-10 dataset [37]. The NeuTRENO model significantly outperforms the in terms of accuracy.

| Model/Metric | Accuracy (%) |
|---|---|
| *Kernel Transformer* | 75.89 |
| NeuTRENO | **76.75** |

# F Additional Experimental Results

## F.1 Object classification on Imagenet with DeiT-small baseline

In this section, we show the advantages of our method when we further scale up the model by doubling the model dimension and the number of heads compared to that of the DeiT-tiny. In particular, the NeuTRENO DeiT-small achieves better results in both Top-1 Accuracy and Top-5 Accuracy, as shown in Table 7. Our method also outperforms DeiT plus FeatScale. Here, we did our best to reproduce the results of DeiT-small plus FeatScale [65]. In Table 7, we include our reproduced results and the results reported in [59] for DeiT-small and [65] for DeiT-small plus FeatScale, respectively.

## F.2 Beyond Softmax-Attention

We show that NeuTRENO can be combined with other baseline attention mechanisms other than softmax attention. In particular, our NeuTRENO significantly improves transformer-based models with kernel attention [54, 61], on the CIFAR-10 image classification task [37], as shown in Table 8. This further confirms the benefits of our model. Here, both models share the same configuration regarding training, the model's size, and the model's depth (12 layers).

# G Additional Empirical Analysis Results

This section provides extra empirical analysis to further demonstrate the benefits of NeuTRENO models in mitigating over-smoothing.

## G.1 Visualizing Attention Matrices

Fig. 4 displays the 3-head attention matrices obtained from layer $[1, 6, 12]$ of both the pre-trained NeuTRENO DeiT-tiny and the DeiT-tiny baseline models, using a random sample from the ImageNet dataset.

## G.2 Head Redundancy between Layers

NeuTRENO mitigates head redundancy between layers, particularly in the final transformer layers where over-smoothing is most pronounced. Fig. 5 shows the average cosine similarity of attention matrices between two successive layers, over 1000 randomly sampled data. The trained NeuTRENO DeiT obtains lower cosine similarity than that of the trained DeiT as the model depth increases.

## G.3 NeuTRENO Inherently Mitigates Over-smoothing, even without Training the Models

Randomly-initialized NeuTRENO DeiT-tiny significantly reduces the average cosine similarity between token representations of 12-layer randomly-initialized DeiT-tiny model, as shown in Fig. 6, on the Imagenet classification task. This observation highlights the ability of our NeuTRENO models in mitigating over-smoothing.

## G.4 Efficiency Analysis

We report the ratios of the floating-point operations per second (FLOPs), the inference memory, and the inference real-time running of NeuTRENO DeiT vs. DeiT per sample on the ImageNet dataset, which are $1.00005, 1.000002, 1.00013$, respectively. This indicates that the significant gain in the performance of NeuTRENO does not come with the cost of efficiency.

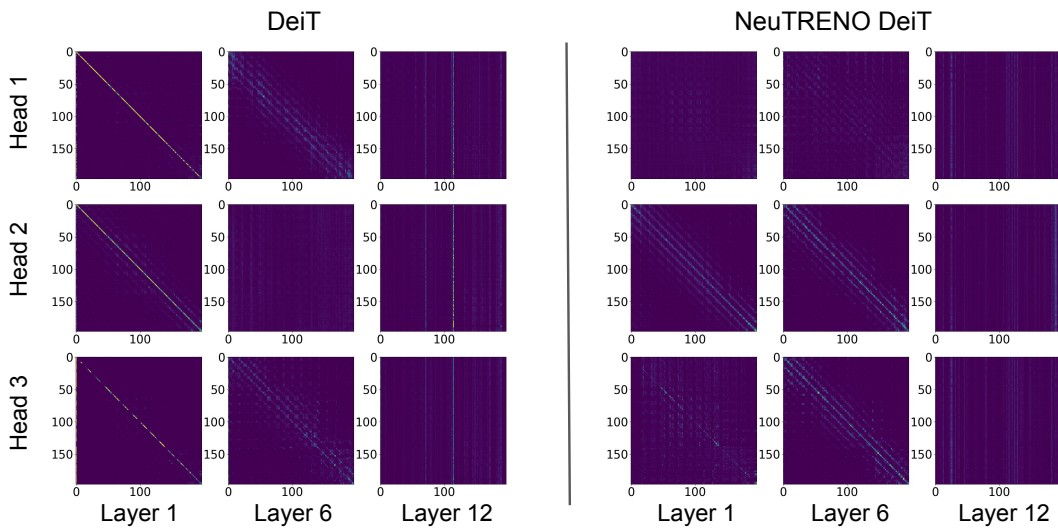

Figure 4: Plot of attention matrices attained from layer $[1, 6, 12]$ of both the pretrained DeiT-tiny baseline (Left) and the NeuTRENO DeiT-tiny (Right) models, for each head, using a random sample from the Imagenet dataset.

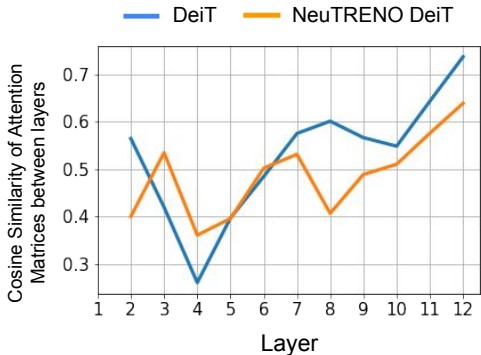

Figure 5: The average cosine similarity of attention matrices between two successive layers, over 1000 randomly sampled data, of the trained NeuTRENO DeiT and trained DeiT models on the Imagenet classification task.

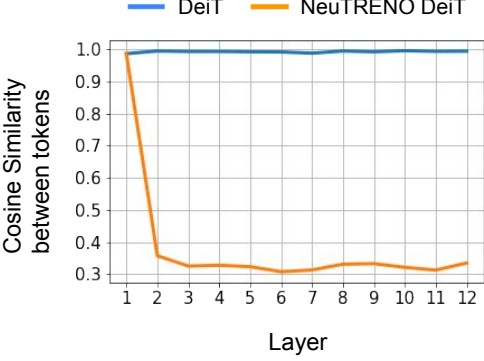

Figure 6: The average cosine similarity between token representations of 12-layer randomly-initialized NeuTRENO DeiT and DeiT models, on the Imagenet classification task. Here, 1000 data are randomly sampled for the analysis.

### G.5 Stability and Significance of NeuTRENO

To further confirm the stability and significance of NeuTRENO's performance, we provide the standard deviations from five runs for both the NeuTRENO and baseline models for each experiment (in the main text) in Tables 9, 10, 11.

Table 9: Means and standard deviations over five runs with different random seeds of models trained on the Imagenet Classification task.

| Model/Metric | Top-1 Acc (%) | Top-5 Acc (%) |
|---|---|---|
| *Softmax DeiT-Tiny* | $72.17 \pm 0.07$ | $91.02 \pm 0.04$ |
| NeuTRENO DeiT-Tiny | $73.01 \pm 0.09$ | $91.56 \pm 0.05$ |
| NeuTRENO Adaptation | $72.63 \pm 0.07$ | $91.38 \pm 0.03$ |
| DeiT-Tiny + FeatScale | $72.346 \pm 0.06$ | $91.22 \pm 0.04$ |
| NeuTRENO DeiT-Tiny + FeatScale | $\mathbf{73.23 \pm 0.08}$ | $\mathbf{91.73 \pm 0.05}$ |

Table 10: Means and standard deviations over five runs with different random seeds of models trained on the ADE20K image segmentation task

| Metric/Model | *Pretrained Softmax Deit-Tiny* | Pretrained NeuTRENO DeiT-Tiny |
|---|---|---|
| SS MIoU | $35.72 \pm 0.57$ | $\mathbf{37.24 \pm 0.62}$ |
| MS MIoU | $36.68 \pm 0.42$ | $\mathbf{38.06 \pm 0.54}$ |

Table 11: Means and standard deviations over five runs with different random seeds of models trained on the WikiText-103 language model task.

| Metric/Model | *Softmax Transformer* | NeuTRENO |
|---|---|---|
| Valid PPL | $33.15 \pm 0.07$ | $\mathbf{32.60 \pm 0.08}$ |
| Test PPL | $34.29 \pm 0.09$ | $\mathbf{33.70 \pm 0.07}$ |

Table 12: Evaluation of NeuTRENO DeiT-Tiny vs. Softmax DeiT-Tiny on the ImageNet-C (mean corruption error mCE), Imagenet-A (Accuracy), and Imagenet-R (Accuracy) datasets.

| Model/Dataset (Metric) | Imagenet-C (mCE) | Imagenet-A (Accuracy %) | Imagenet-R (Accuracy %) |
|---|---|---|---|
| *Softmax DeiT-Tiny* | 71.6 | 6.9 | 32.83 |
| NeuTRENO-DeiT-Tiny | **70.1** | **8.2** | **33.82** |

Table 13: Ablation study of different values hyperparameter $\tilde{\lambda}$ of NeuTRENO DeiT-Tiny on the ADE20K Image Segmentation task.

| Metric/Model | *Baseline* | $\tilde{\lambda} = 0.1$ | $\tilde{\lambda} = 0.2$ | $\tilde{\lambda} = 0.4$ | $\tilde{\lambda} = 0.5$ | $\tilde{\lambda} = 0.6$ | $\tilde{\lambda} = 0.8$ | $\tilde{\lambda} = 1.0$ | $\tilde{\lambda} = 2.0$ |
|---|---|---|---|---|---|---|---|---|---|
| SS MIoU | 35.72 | 34.94 | 35.60 | **37.54** | 37.38 | 37.24 | 36.71 | 36.37 | 26.25 |
| MS MIoU | 36.68 | 35.53 | 36.45 | **38.62** | 38.22 | 38.06 | 37.82 | 37.26 | 27.2 |

## G.6 Robustness of NeuTRENO

In addition to the standard metrics, we evaluate the robustness of our NeuTRENO model compared to the baseline transformer model, particularly under adversarial examples and for out-of-distribution generalization. Table 12 demonstrates that NeuTRENO DeiT-Tiny is consistently more robust than the DeiT-Tiny baseline on the Imagenet-C (common data corruption and perturbations, such as adding noise and blurring the images) [29], Imagenet-A (adversarial examples) [30], and Imagenet-R (out of distribution generalization) [28] datasets, which are widely used to test the model's robustness.

## G.7 NeuTRENO in Incremental Learning

In an incremental learning setting [34], our 8-layer NeuTRENO achieves 1.97% higher accuracy on the sentiment classification task [36] than the 8-layer baseline transformer.

## G.8 Ablation study on the choice of $\tilde{\lambda}$

We also conduct an ablation study on the impact of the hyperparameter. In particular, on the ADE20K image segmentation task, we train NeuTRENO with different values. We summarize our results in Table 13. Our findings reveal that within the range of $[0.2, 1]$, NeuTRENO consistently outperforms the softmax baseline. However, when values become small or big (below 0.2 or above 1, respectively), NeuTRENO's performance declines.

## G.9 Scalability of NeuTRENO

To demonstrate the scalability of our proposed model, we conduct additional experiments to show that our NeuTRENO method can effectively mitigate the oversmoothing issue in the BERT-base model. In particular, in Figure 7 (Left), we plot the cosine similarity between token representations across layers of a pre-trained BERT-base model [17] on the SQuAD v1.1 question answering task [48] and observe

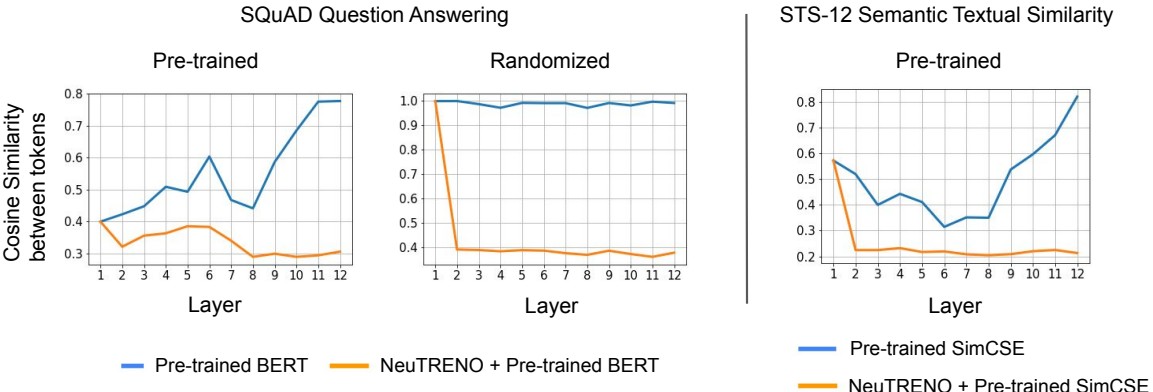

Figure 7: The average cosine similarity between token representations of 12-layer trained (Left) and randomly-initialized (Middle) BERT-base and NeuTRENO BERT-base on the SQuAD question answering task. We also plot the same cosine similarity scores for the trained SimCSE and NeuTRENO SimCSE models (Right) on the STS-12 semantic textual similarity task. Here, 1000 and 500 data are randomly sampled for the analysis on the SQuAD and STS-12 datasets, respectively.

the presence of the oversmoothing issue as the model gets deeper, causing tokens to become identical. We then apply NeuTRENO on the same pre-trained BERT model, and without any fine-tuning, we observe a significant reduction in the cosine similarity between token embeddings in each layer (see Figure 7 (Left)), indicating that NeuTRENO effectively mitigates the oversmoothing problem in BERT. Additionally, our NeuTRENO BERT finetuned on the task yields better accuracy than the finetuned BERT (81.39 exact match score and 88.62 F1-score vs. 80.77 exact match score and 88.12 F1-score). Moreover, we have conducted the same analysis for a randomized BERT-base model and a randomized NeuTRENO BERT-base model and obtained the same encouraging results (see Figure 7 (Middle)). These results further suggest that NeuTRENO helps alleviate the over-smoothing issue in large-scale transformer models.

We also obtain additional results and show that our NeuTRENO SimCSE, after fine-tuned on the STS-12 semantic textual similarity task [1], gains a significant improvement over the baseline SimCSE [22], which is also fine-tuned on the same task (77.32% vs. 75.29% Spearman's correlation. Here, the higher correlation, the better). This additional result further verifies that decreasing the cosine dissimilarity between tokens within trained transformer-based models leads to improved empirical performance.

