# OpenReview forum: "Mitigating Over-smoothing in Transformers via Regularized Nonlocal Functionals"
_NeurIPS.cc/2023/Conference — NeurIPS 2023 poster_

### Official Review · Reviewer_k2Mg · 2023-06-17

**Soundness:** 4 excellent
**Presentation:** 3 good
**Contribution:** 4 excellent
**Rating:** 7
**Confidence:** 3

**Summary:**

Recent research works have revealed that the over-smoothing issue, a prevalent challenge in Graph Neural Networks, similarly plagues Transformers. Contrary to expectations, the performance of a Transformer model does not invariably improve with increased depth of the self-attention layers. In fact, a deeply layered Transformer may not necessarily outperform a shallow one. This paper elucidates that the over-smoothing phenomenon occurs due to a consistent decrease in the non-local function of weighted differences in each self-attention layer as depth increases. Drawing on this finding, the paper puts forward a solution featuring modified skip connections. While similar solutions have been proposed in prior research, this paper sets itself apart by providing an in-depth theoretical analysis from a gradient flow perspective, which previous works have not addressed. Furthermore, the experimental results corroborate the effectiveness of the proposed solution in mitigating over-smoothing in Transformers.

**Strengths:**

1. The theoretical investigation conducted in this paper is rigor and in-depth.

**Weaknesses:**

1. The theoretical explication put forth in this paper applies accurately only to Transformers with single-head self-attention mechanisms. However, given that the prevailing Transformers utilize multi-head self-attention mechanisms and feed-forward networks, it becomes prudent to critically evaluate the efficacy of the resolution proposed herein.
2. This paper lacks a comprehensive set of comparative experiments to analyze the influence of diverse initial values of the parameter $\widetilde{\lambda}$ on the model's final performance.

**Questions:**

1. This paper introduces a concept of symmetric self-attention, i.e., the query matrix is equal to the key matrix. The experimental results demonstrate an unexpectedly high performance from a Transformer predicated on symmetric self-attentions. This performance seems to contravene the conventional understanding that self-attention should capture the bi-directional correlation among tokens. Can the authors provide a logical elucidation to account for these particular experimental outcomes?

---

> ### Author Rebuttal · Authors · 2023-08-09
>
>
> Thank you for your thoughtful review and valuable feedback. Below we address your concerns.
>
> **Q1**. The theoretical explication put forth in this paper applies accurately only to Transformers with single-head self-attention mechanisms. However, given that the prevailing Transformers utilize multi-head self-attention mechanisms and feed-forward networks, it becomes prudent to critically evaluate the efficacy of the resolution proposed herein.
>
> **Reply:** Thanks for your comments. The variational denoising framework we develop for the single-head self-attention mechanism can be extended to derive the multi-head self-attention mechanism. Please allow us to explain this derivation below.
>
> Let us consider one of the common implementations of the multi-head self-attention, in which the input sequence $X \in R^{N \times D_{x}}$ is truncated into $H$ pieces $X_d \in R^{N \times D_{x}/H}$, for $d = 1, ..., H$. Here, we assume that $D_{x}$ is divisible by $H$.  In order to derive the multi-head attention, we apply our variational denoising framework proposed in Section 2 and Theorem 1 in our paper on each of these $X_d$ signals. Note that the vector $\bf{q}$$(i)$, $\bf{k}$$(i)$, and $\bf{v}$$(i)$, for $i = 1,...,N$,  in Theorem 1 are replaced by the vectors $\bf{q}$$_d(i)$, $\bf{k}$$_d(i)$, and $\bf{v}$$_d(i)$ which are linear transformations of the input vector $X_d(i)$, $i = 1,...,N$. The feed-forward network then combines the output sequences of all of these $H$ heads.
>
> Given our derivation above, it follows that the oversmoothing issue still persists in transformers with multi-head self-attention, and the proposed NeuTRENO still helps mitigate it. Indeed, our experiments in the paper are conducted with multi-head self-attention and validate the advantage of NeuTRENO over the baseline multi-head self-attention in resolving oversmoothing.
>
> **Q2**. This paper lacks a comprehensive set of comparative experiments to analyze the influence of diverse initial values of the parameter on the model's final performance.
>
> **Reply:** Thank you for your suggestion. We have conducted an ablation study on the impact of the hyperparameter $\tilde\lambda$. In particular, on the ADE20K image segmentation task, we train NeuTRENO with different $\tilde\lambda$ values. We summarize our results in Table 13 in the attached PDF. Our findings reveals that within the range of [0.2, 1], NeuTRENO consistently outperforms the softmax baseline. However, when $\tilde\lambda$ values become small or big (below 0.2 or above 1, respectively), NeuTRENO's performance declines.
>
> **Q3**. This paper introduces a concept of symmetric self-attention, i.e., the query matrix is equal to the key matrix. The experimental results demonstrate an unexpectedly high performance from a Transformer predicated on symmetric self-attentions. This performance seems to contravene the conventional understanding that self-attention should capture the bi-directional correlation among tokens. Can the authors provide a logical elucidation to account for these particular experimental outcomes?
>
> **Reply:** Thank you for your question. First, we would like to clarify that transformer models can be bi-directional or uni-directional, depending on the task or model design. For instance, BERT [1], a bi-directional model, allows tokens to attend to both preceding and succeeding tokens. On the other hand, models like GPT [2], trained for autoregressive language modeling, is a uni-directional model as tokens can only attend to their previous counterparts. Second, regarding the symmetric attention, its competitive empirical performances have been demonstrated in existing works [3, 4].
>
> **References**
>
> [1] Jacob Devlin, et al.. “BERT: Pre-training of Deep Bidirectional Transformers for Language Understanding”. NAACL, 2019.
>
> [2] Tom Brown, et al. "Language models are few-shot learners." NEURIPS, 2020.
>
> [3] Yao-Hung Hubert Tsai,et al. “Transformer Dissection: An Unified Understanding for Transformer’s Attention via the Lens of Kernel”. EMNLP-IJCNLP, 2019.
>
> [4] Wenlong Chen, et al. "Calibrating Transformers via Sparse Gaussian Processes." ICLR, 2022.
>
> -----
> We hope we have cleared your concerns about our work. We would appreciate it if we could get your further feedback at your earliest convenience.

---

> > ### Comment · Reviewer_k2Mg · 2023-08-21
> >
> > Thanks for the authors' response. I have no further questions about this paper.

---

> ### Author Response · Authors · 2023-08-20
> **Thanks for your endorsement!**
>
> Thanks for increasing your score, and we appreciate your endorsement.

---

### Official Review · Reviewer_qcqa · 2023-07-03

**Soundness:** 3 good
**Presentation:** 2 fair
**Contribution:** 3 good
**Rating:** 6
**Confidence:** 3

**Summary:**

This work shows that self-attention layers in transformers minimize a funcitonal which promotes smoothneess, thereby casuing token uniformity. The work also proposes a novel regularizer to preserve fidelity of the tokens. The work empirically shows that NeuTRENO outperforms baseline transformers in reducing the over-smoothing of token representation.

**Strengths:**

- I like the flow of the paper. The authors dive straight into the key issue of the work, without much retrospect on the previous known matters.
- The paper rewrites the self-attention as a Gradient Descent Step to minimizie a nonlocal functional.
- The paper shows that as k (number of layers) increases, the model is more likely to suffer from over-smoothing.
- I think viewing each self-attention layer in the transformer as an impliciit gradient ascent in very interesting. It echos with some literature in the past, providing more insights on the interpretability of deep neural networks.

**Weaknesses:**

- Some use of the wording could be improved, for example in line 220s, the authors like to use words like "significantly", "addresses" etc. In my opinion, some of the experimental results are not strong enough to support author's claim. It could create misleading for the audiences.
- Experiments looks very interesting. I think the paper might benefit more by discussing broader impact of "alleviating over-smoothness". Say for example. would be help in OOD setting? incremental setting? or other interesting applications. In my opinion, 1% increase on standard metric should not be the best selling point for this interesting work.

**Questions:**

- Although the authors motivates this work from the point of over-smoothing, the paper lacks discussion on the actual cause, impact and consequence of over-smoothing. For example, from Figure.1, we observe that NeuTRENO has a smaller cosine similarity comparing to naive DeiT. One question would be, what should be the desired cosine sim? Is smaller cosine sim siginificantly better than say sim over 0.8?

**Limitations:**

Please see the previous two sections

---

> ### Author Rebuttal · Authors · 2023-08-09
>
> Thank you for your thoughtful review and valuable feedback. Below we address your concerns.
>
> **Q1**. Some use of the wording could be improved, for example in line 220s, the authors like to use words like "significantly", "addresses" etc. In my opinion, some of the experimental results are not strong enough to support author's claim. It could create misleading for the audiences.
>
> **Reply:** Thank you for your feedback. We have removed those words in our revised manuscript. We have also included the standard deviations from 5 runs for each experiment (in the main text) in Tables 9, 10, and 11 in the attached PDF. The standard deviations for both the NeuTRENO and baseline models in these tasks are small compared to the gains achieved by the NeuTRENO method. This observation suggests that NeuTRENO’s  improvements over the baseline are significant and not by chance.
>
> **Q2**. Experiments looks very interesting. I think the paper might benefit more by discussing broader impact of "alleviating over-smoothness". Say for example. would be help in OOD setting? incremental setting? or other interesting applications. In my opinion, 1% increase on standard metric should not be the best selling point for this interesting work.
>
> **Reply:** Thank you for your valuable suggestions.  We have evaluated the robustness of our NeuTRENO model compared to the baseline transformer model, particularly under adversarial examples and for out-of-distribution generalization. Table 12 in the attached PDF demonstrates that NeuTRENO DeiT-Tiny is consistently more robust than the DeiT-Tiny baseline on the Imagenet-C ( corruption and perturbations data, such as adding noise and blurring the images) [1], Imagenet-A (adversarial examples) [2], and Imagenet-R (for out of distribution generalization) [3], which are widely used to test the model’s robustness.
>
> Furthermore, in an incremental learning setting [4], our 8-layer NeuTRENO achieves 1.97% higher accuracy on the sentiment classification task [5] than the 8-layer baseline transformer. We hope that these additional analyses and results address your concerns and provide further evidence for the stability, significance, and robustness of our proposed NeuTRENO approach.
>
> **References**
>
> [1] Hendrycks, Dan, et al. "Benchmarking neural network robustness to common corruptions and perturbations." arXiv, 2019.
>
> [2] Hendrycks, Dan, et al. "Natural adversarial examples." CVPR, 2021.
>
> [3] Hendrycks, Dan, et al. "The many faces of robustness: A critical analysis of out-of-distribution generalization." ICCV, 2021.
>
> [4] Kahardipraja, et al. "Towards incremental transformers: An empirical analysis of transformer models for incremental NLU."  EMNLP, 2021.
>
> [5] Dimitrios Kotzias, et al. "From group to individual labels using deep features". KDD, 2015
>
> **Q3**. Although the authors motivates this work from the point of over-smoothing, the paper lacks discussion on the actual cause, impact and consequence of over-smoothing. For example, from Figure.1, we observe that NeuTRENO has a smaller cosine similarity comparing to naive DeiT. One question would be, what should be the desired cosine sim? Is smaller cosine sim siginificantly better than say sim over 0.8?
>
> **Reply:** Thanks for your comments. We believe there is a misunderstanding of the contributions of our paper. Please allow us to clear this misunderstanding by clarifying that our work first focuses on developing a variational denoising framework to understand the self-attention of transformers as a gradient descent approximation of a functional. Using this new finding, we provide an explanation for the oversmoothing issue in transformers as a result of self-attention minimizing a functional, leading to the smoothing effect on the input sequence, analogous to a diffusion process  (see Remark 1 in our main text). Then, we  rigorously prove this observation on the existence of oversmoothing in transformers using a random walk analysis in Section 2.2. Thus, in our paper, we not only discuss the actual cause of oversmoothing but also prove its existence theoretically.
>
> Regarding the impact and consequence of oversmoothing, oversmoothing leads to a loss of diversity among token representations, hindering the model's capacity to capture diverse features. As a result, the model's performance can be adversely affected. Additionally, oversmoothing often constrains the ability of transformer models to be scaled up in depth. For instance, more-layer transformer models might underperform compared to those with fewer layers due to this issue [1].
>
> The cosine similarity between token representations of each layer is an indicator of the oversmoothing issue. It shows how similar a token is to the other, on average, at a layer. When cosine similarity at a layer is high, it indicates that the representations of tokens are too similar, diminishing the expressive power of the model. Consequently, smaller cosine similarity scores are preferable.
>
> **References**
>
> [1] Daquan Zhou, et al. “Deepvit: Towards deeper vision transformer”. arXiv, 2021.
>
> -----
> We hope we have cleared your concerns about our work. We would appreciate it if we could get your further feedback at your earliest convenience.

---

> > ### Comment · Reviewer_qcqa · 2023-08-16
> >
> > I thank the author for providing this rebuttal. I agree that it is a very interesting work that tried to understand the self-attention of transformers as a gradient descent step. I am still a bit confused by the over-smoothness sections. For example, in figure.7 of the rebuttal pdf, the proposed new architecture seem to successfully decrease the cosine sim between tokens. But does that lead to better empirical performance or better interpretability? On vision tasks, I am still dubious about the statement that over-smooth is indeed a issue and should be addressed. Of course, I think the work is very valuable in terms of trying to address this problem if this problem is indeed very significant.

---

> > > ### Author Response · Authors · 2023-08-17
> > > **Response to Reviewer qcqa: Oversmoothing and Empirical Performance**
> > >
> > >
> > > Thanks for your further feedback. Please allow us to address your concerns below. We will include the following discussion and existing works in our revised manuscript to clarify the impact of oversmoothing on the model’s performance, especially for the vision transformer (ViT).
> > >
> > > **Question: For example, in figure.7 of the rebuttal pdf, the proposed new architecture seems to successfully decrease the cosine sim between tokens. But does that lead to better empirical performance or better interpretability?**
> > >
> > > **Reply:** In Figure 7 (Left) of the rebuttal PDF, our NeuTRENO BERT finetuned on the SQUAD question answering task [1] yields better accuracy than the baseline BERT finetuned on the same task (81.39 exact match score and 88.62 F1 score vs. 80.77 exact match score and 88.12 F1 score). This result indicates that reducing cosine similarity between tokens in the trained transformer-based models leads to better empirical performance.
> > >
> > > **Question: On vision tasks, I am still dubious about the statement that over-smooth is indeed an issue and should be addressed.**
> > >
> > > **Reply:** The oversmoothing in ViT has been verified and investigated in existing works. In particular, [2] observes that the performance of ViT quickly saturates as more layers are added to the model. Moreover, experiments in [2] show that the 32-layer ViT underperforms the 24-layer ViT, indicating the difficulty of ViTs in gaining benefits from deeper architectures. The authors point out that oversmoothing results in this phenomenon by causing the token representations to become identical when the model grows deeper. Based on this observation, they propose a cross-head communication method that helps enhance the diversity of both token representations and attention matrices.
> > >
> > > Furthermore, it has been shown in [3] that the training of ViT models encounters instability with greater depths. [4] proposes that this instability arises from the oversmoothing, where token representation for patches within an image become progressively alike as the model's depth increases. In an effort to explain this issue, [5] finds out that self-attention acts as a low-pass filter, which smoothens the token representations in ViTs. This leads to the proposal of the FeatScale method [5], which regulates feature frequencies, whether low or high, to counteract the consequences of oversmoothing.
> > >
> > > Different from existing works, in our paper, we theoretically prove the oversmoothing phenomenon via the variational denoising framework that we develop. Our proposed NeuTRENO method helps mitigate the oversmoothing issue and improves the performance of the ViT baselines (DeiTs), as well as other baseline transformer-based models. We also empirically demonstrate that NeuTRENO not only reduces the cosine similarity between token representation (See Figure 1 in the main text and Figure 7) but also the redundancy in attention maps between layers (See Figure 5 in Appendix G2). Finally, we show that NeuTRENO is complementary to existing methods, including FeatScale [5] (See Table 1 and Table 7  in our paper).
> > >
> > > **References**
> > >
> > > [1] Pranav Rajpurkar, Jian Zhang, Konstantin Lopyrev, and Percy Liang. “SQuAD: 100,000+ questions for machine comprehension of text”. EMNLP, 2016.
> > >
> > > [2] Daquan Zhou, et al. “Deepvit: Towards deeper vision transformer”. arXiv, 2021.
> > >
> > > [3] Touvron Hugo, et al. "Going deeper with image transformers." ICCV, 2021.
> > >
> > > [4] Chengyue Gong, et al. “Vision transformers with patch diversification”. arXiv, 2021.
> > >
> > > [5] Wang Peihao, et al. "Anti-oversmoothing in deep vision transformers via the fourier domain analysis: From theory to practice." ICLR, 2022.

---

> > > > ### Author Response · Authors · 2023-08-19
> > > > **Response to Reviewer qcqa: Oversmoothing and Empirical Performance (Continued with Additional Results)**
> > > >
> > > > Following your suggestion, we have obtained additional results to further address your concern about the impact of oversmoothing on the model's performance. In particular, in Figure 7 (Right) in the rebuttal PDF, our NeuTRENO SimCSE, after fine-tuned on the STS-12 semantic textual similarity task [1], gains a significant improvement over the baseline SimCSE [2], which is also fine-tuned on the same task (77.32% vs. 75.29% Spearman's correlation. Here, the higher correlation, the better). This additional result further verifies that decreasing the cosine dissimilarity between tokens within trained transformer-based models leads to improved empirical performance.
> > > >
> > > > **References**
> > > >
> > > > [1] Eneko Agirre, et al. "SemEval-2012 task 6: A pilot on semantic textual similarity." SemEval, 2012.
> > > >
> > > > [2] Gao, T., Yao, X., and Chen, D. "SimCSE: Simple Contrastive Learning of Sentence Embeddings." EMNLP, 2021.

---

> > > > > ### Comment · Reviewer_qcqa · 2023-08-20
> > > > >
> > > > > Thank the authors for providing these additional clarifications. Overall the contribution in this paper is non-trivial and very interesting. I agree that adding the clarifications in the rebuttal in the paper would strengthen the work. (Also the effort to make such changes would also be non-trivial lol). Given the responses, I encourage the authors explore tasks mentioned in the rebuttal (e.g. training deeper networks) with this new architectur for future works.

---

> > > > > > ### Author Response · Authors · 2023-08-20
> > > > > > **Response to Reviewer qcqa: Thanks for your endorsement!**
> > > > > >
> > > > > > Thanks for your valuable feedback. We are encouraged by your endorsements that our paper is non-trivial and very interesting. We will add the clarifications on the oversmoothing phenomenon in vision transformers and our results during the rebuttal, which show that reducing oversmoothing helps improve the model’s performance, into the revision. Following your suggestion, we will also include our new results for training NeuTRENO under adversarial examples and for out-of-distribution generalization, as well as for the incremental learning task, in the revised manuscript. We will further study the applications of NeuTRENO on these tasks and train deeper NeuTRENO in future works.

---

### Official Review · Reviewer_h3ak · 2023-07-06

**Soundness:** 3 good
**Presentation:** 3 good
**Contribution:** 3 good
**Rating:** 6
**Confidence:** 4

**Summary:**

This paper analyses the over-smoothing problem of transformer architecture by showing that self-attention layers minimize a functional that causes over-smooth. To address this problem, the authors introduce a regularizer that penalizes the norm of the difference between the smooth output tokens and input tokens. In the experimental section, the authors demonstrate that their proposed solution, NeuTRENO, outperforms baseline models on visual and language-related downstream tasks.

**Strengths:**

The issue of over-smoothing in transformers is a fascinating topic. While previous research has explored this problem, this paper offers a unique perspective by examining self-attention as a gradient descent step from a variational standpoint. This novel approach sheds new light on the over-smoothing problem and contributes to a deeper understanding of the issue within the research community.

The effectiveness of the proposed solution, NeoTRENO, in mitigating the problem of over-smoothing is demonstrated both theoretically and empirically. The experiments show NeoTRENO's effectiveness in both vision and language transformers.

**Weaknesses:**

Current experiments are mostly conducted on small transformer models( e.g., DeiT-tiny).  Given that the transformer architecture is crucial for large pre-trained language models, it remains unclear whether the proposed solution can effectively alleviate the over-smoothing problem when the model size is increased, and whether can be combined with large pre-trained language models (e.g.,  BERT).



**Questions:**

please refer to weaknesses

**Limitations:**

The generalizability and scalability of the proposed solution to larger transformer models have not been investigated.

---

> ### Author Rebuttal · Authors · 2023-08-09
>
> Thank you for your thoughtful review and valuable feedback. Below we address your concerns.
>
> **Q1.** Current experiments are mostly conducted on small transformer models( e.g., DeiT-tiny). Given that the transformer architecture is crucial for large pre-trained language models, it remains unclear whether the proposed solution can effectively alleviate the over-smoothing problem when the model size is increased, and whether can be combined with large pre-trained language models (e.g., BERT).
>
> **Reply:** Thanks for your comments. The DeiT-tiny model used in our experiments has 5M parameters and is trained on the large-scale Imagenet dataset for classification task. To examine the scalability of our NeuTRENO model, we have applied NeuTRENO to the DeiT-small, which is 4 times larger than the DeiT-tiny and has 22M parameters. We summarize the results in Table 7 in Appendix F.1. The improvements achieved by our NeuTRENO DeiT-small over the baseline model demonstrate the scalability of NeuTRENO when applied to very large transformer architectures.
>
> Furthermore, we have conducted additional experiments to show that our NeuTRENO method can effectively mitigate the oversmoothing issue in the BERT-base model. In particular, in Figure 7 (Left) in the attached PDF, we plot the cosine similarity between token representations across layers of a pre-trained BERT-base model [1] on the SQuAD v1.1 question answering task [2] and observe the presence of the oversmoothing issue as the model gets deeper, causing tokens to become identical. We then apply NeuTRENO on the same pre-trained BERT model, and without any fine-tuning, we observe a significant reduction in the cosine similarity between token embeddings in each layer (see Figure 7 (Left) in the attached PDF), indicating that NeuTRENO effectively mitigates the oversmoothing problem in BERT. Moreover, we have conducted the same analysis for a randomized BERT-base model and a randomized NeuTRENO BERT-base model and obtained the same encouraging results (see Figure 7 (Middle) in the attached PDF). These results further suggest that NeuTRENO helps alleviate the oversmoothing issue in large-scale transformer models.
>
> **References**
>
> [1] Jacob Devlin, et al. “BERT: Pre-training of Deep Bidirectional Transformers for Language Understanding”. NAACL, 2019.
>
> [2] Pranav Rajpurkar, et al. “SQuAD: 100,000+ Questions for Machine Comprehension of Text”. EMNLP, 2016.
>
> -----
> We hope we have cleared your concerns about our work. We would appreciate it if we could get your further feedback at your earliest convenience.

---

> > ### Comment · Reviewer_h3ak · 2023-08-18
> >
> > Thanks for the authors' response and clarifiaction. I keep my initial rating.

---

> > > ### Author Response · Authors · 2023-08-18
> > > **Thanks for your endorsement!**
> > >
> > > Thanks for your response and we appreciate your endorsement.

---

### Official Review · Reviewer_YNWk · 2023-07-07

**Soundness:** 3 good
**Presentation:** 3 good
**Contribution:** 3 good
**Rating:** 5
**Confidence:** 4

**Summary:**

This paper studies the oversmoothing problem in transformers. Roughly speaking, it was observed that embeddings start to converge when the network is deeper. The authors built a model to explain this phenomenon. Roughly speaking, they relate having deeper architecture with making progress towards minimizing a function. Then they built some regularizers out of this intuition.

**Strengths:**


I think these analysis and regularization tools are useful addition to the exploding area of transformers.


**Weaknesses:**


I feel I was not able to properly parse some math texdt. My most unsure part is whether the amount of hand-waiving is too much even under today’s standard. For example, the authors feel very comfortable in exchanging integration operator and summation. It is true in general the approximation errors are usually inconsequential. But it is still different from pretending that they are the same in theorem statements/proofs. Also, I dont understand how K(x, y), k(x, y) and boldk(x) and boldk(y) are related. First k(x, y) are defined, then there will always exist boldk? It also looks that sometimes k/K are treated as kernels whereas sometimes it is treated as a kernel function. Probability functions and kernel functions have different properties?


**Questions:**

It was proven that an update of a self-attention is equivalent to taking a gradient over a function. But will the function have many local optimal so that even we have the main theorem, it does not quite say all the embeddings will eventually be the same? Or maybe the function is a convex function.

---

> ### Author Rebuttal · Authors · 2023-08-09
>
> Thank you for your thoughtful review and valuable feedback. Below we address your concerns.
>
> **Q1**. I feel I was not able to properly parse some math texdt. My most unsure part is whether the amount of hand-waiving is too much even under today’s standard. For example, the authors feel very comfortable in exchanging integration operator and summation. It is true in general the approximation errors are usually inconsequential. But it is still different from pretending that they are the same in theorem statements/proofs.
>
> **Reply:** In the main text, we employ the Monte Carlo approximation method to estimate integrals in Eqns. 11, 15, 23, and 27. We choose the Monte Carlo integration method because it is an unbiased estimator. Additionally,  when the sample size $N$ used in Monte Carlo is sufficiently large, the estimation converges to the correct value. This tends to hold true for transformer models, which are often applied on long sequences with many token samples.
>
> **Q2**. Also, I don't understand how K(x, y), k(x, y) and boldk(x) and boldk(y) are related. First k(x, y) are defined, then there will always exist boldk? It also looks that sometimes k/K are treated as kernels whereas sometimes it is treated as a kernel function. Probability functions and kernel functions have different properties?
>
> **Reply:** The function $k(x, y)$ captures the similarity between signal values at positions x and y. For example, $k(x, y)$ can be a radial basis function (RBF) kernel $e^{-\gamma||\Phi(x) - \Phi(y)||^2}$ for a feature function $\Phi$ [1]. $K(x, y)$ is indeed just a shorthand notation for $k(x, y) + k(y, x)$. We use this notation to simplify the expression.
>
> $\bf{k}$$(\cdot)$ is a vector-valued function which maps positions ($x$ or $y$) to feature vectors. In transformer models, ${\bf k}(i)$, $i=1,\dots,N$, is the key vector of token $i$-th in self-attention. For images, $\bf{k}$$(x)$ indicates the feature vector at pixel $x$.
>
> **References**
>
> [1] Bishop, Christopher M. "Pattern Recognition and Machine Learning". New York. Springer, 2006.
>
> **Q3. It was proven that an update of a self-attention is equivalent to taking a gradient over a function. But will the function have many local optimal so that even we have the main theorem, it does not quite say all the embeddings will eventually be the same? Or maybe the function is a convex function.**
>
> **Reply:** Yes, the functional $E$ is a convex function of $\bf{u}$ since $J(\bf{u})$ is a sum of quadratic functionals of $u_j$ for $j = 1,\dots, D$, and $G(\bf{u}, \bf{f})$ is a quadratic functional of $\bf{u}$. Hence, $E(\bf{u})$, the sum of two convex functional, is itself a convex functional.
>
> -----
> We hope we have cleared your concerns about our work. We would appreciate it if we could get your further feedback at your earliest convenience.

---

> > ### Author Response · Authors · 2023-08-22
> > **Thank You**
> >
> > Dear Reviewer YNWk,
> >
> > Thanks for your reviews of our paper. Since the discussion period between the authors and the reviewers was already over and we have not heard from you during this period, we would be grateful if the reviewer could let us know if all your questions are addressed to some extent. If you are satisfied with our answers, we hope that the reviewer will consider adjusting your score.
> >
> > Best regards,
> >
> > Authors

---

### Official Review · Reviewer_qvvE · 2023-07-10

**Soundness:** 2 fair
**Presentation:** 2 fair
**Contribution:** 2 fair
**Rating:** 4
**Confidence:** 5

**Summary:**

This paper analyzes the reason of over-smoothing in Transformer based on Self-attention as a Gradient Descent Step to Minimize a Nonlocal Function and random walk analysis. The paper then proposes a novel regularizer that penalizes the norm of the difference between the output tokens from self-attention and the input tokens to preserve the fidelity of  the tokens.  Experimental results on ImageNet classification, image segmentation and LM tasks show that the proposed approach achieves better performance than the baseline vanilla Transformer.

**Strengths:**

Strengths:

(1)	The paper provides interesting aspects to analyze the reason of over-smoothing in Transformers based on Self-attention as a Gradient Descent Step to Minimize a Nonlocal Function and random walk analysis.

(2)	The theoretical proofs are detailed and clear.


**Weaknesses:**

Weaknesses:

(1)	Although the paper provides detailed theoretical proofs of the nonlocal variational denoising framework for self-attention and provides an explanation for the over-smoothing issue in transformer-based models, the empirical evaluations missed some important details and analyses.  In Table 1, the configuration of NeuTRENO Adaptation is not explained.

(2)	Except the results in Table 2 on image segmentation, the gains from the proposed approach in Table 1 ImageNet classification and WikiText-103 LM are all quite small. In Table 1, the gains from the proposed approach over the baseline are 0.84 on Top-1 acc and 0.54 on Top-5 acc. In Table 3, the gains from the proposed approach over baseline are 0.55 on validation PPL and 0.59 on test PPL. It is not clear how stable the proposed model is on these datasets, since there is no reporting of standard deviations from multiple runs with different random seeds, so it is not clear whether the gains are by chance. Also, it is not clear whether these gains are statistically significant.


(3)	There have been prior works analyzing the uniformity and alignment problem of BERT on sentence representations and proposed postprocessing solutions, for example, the flow https://arxiv.org/pdf/2011.05864.pdf and BERT whitening method https://arxiv.org/abs/2103.15316 and supervised method such as SimCSE https://arxiv.org/abs/2104.08821 and enhancement over SimCSE on sentence representations.

These works are not compared to or cited in the paper.

(4)	An important point is missing. Based on the results from models addressing uniformity in BERT and improving sentence representations, these models alleviate uniformity but cause degradations in transfer learning capability, e.g., SimCSE. This issue has not been discussed in the paper since the paper did not investigate the proposed approach in pre-training and fine-tuning paradigm, to investigate the impact of the proposed approach on transferability.

(5)	There are some presentation problems. Some math symbols and functions are not defined or clearly defined, as listed under Questions.


**Questions:**

Please address the points under weaknesses.

Also, there are some presentation errors in the paper. For example,
(1)	The word “functional” is an adjective, so using it as a noun in the paper is ungrammatical, for example, in the title and phrases such as “minimize a functional which…” “Minimizing the resulting regularized energy functional, …” throughout the paper.

(2)	Some math symbols have not been defined, for example, D_x , D_{qk} in Section 1.1.

(3)	Line 89 defines the weights k(x,y), but the definition is kind of vague. Line 117 provides a form of K(x,y) :=k(x,y)+k(y,x), which needs to be explained clearer about this choice of K(x,y).


**Limitations:**

This is N/A for this work.

---

> ### Author Rebuttal · Authors · 2023-08-09
>
> Thank you for your thoughtful review and valuable feedback. Below we address your concerns.
>
> **Q1**. The empirical evaluations missed some important details and analyses. In Table 1, the configuration of NeuTRENO Adaptation is not explained.
>
> **Reply:**
> Thank you for your comment. In our NeuTRENO Adaptation experiment, we integrate the NeuTRENO architecture into a pre-trained DeiT-Tiny model with $\tilde\lambda = 0.6$ (determines how much the regularization impacts the solution, a small value makes the solution smoother and vice versa). This combined model is then finetuned for 100 additional epochs. The experiment and its configuration are detailed from lines 250 to 254 (main text) and lines 520 to 523 (Appendix A). For comparisons with related work (e.g., FeatScale method), please refer to lines 234 to 236 (main text). For evaluation metrics, we provide thorough descriptions in Appendix A, under "Datasets and Metrics" for each task.
>
> As for the analysis, we provide numerous empirical analyses of our models in Section 5 (main text) and in Appendix G. In particular, Fig. 1 shows that the cosine similarity between token representations across layers in a trained NeuTRENO is significantly reduced compared to the baseline model, demonstrating its advantage in mitigating oversmoothing. Furthermore, Fig. 6 presents similar results for a randomly initialized NeuTRENO, i.e., before training. Additionally, Fig. 5 (Appendix G2) demonstrates NeuTRENO's advantage in reducing head redundancy compared to the baseline. To address the efficiency of NeuTRENO, an efficiency analysis is provided in Appendix G4. Finally, to verify our theoretical results in Theorem 1, in Figure 3, we show that softmax attention indeed minimizes the functional $J(\bf{u})$ in Eqn. 6 in our paper.
>
> **Q2**. Question on the stability and significance of NeuTRENO's performance.
>
> **Reply:** Thank you for your question. Please kindly refer to points 1, 2, and 3 in the general response for our answers to your questions.
>
> **Q3**. Reviewer's concerns that prior work on the uniformity and alignment problem of BERT on sentence representations are overlooked.
>
> **Reply:** Thanks for bringing up these prior works. However, we respectfully disagree that our paper disregards these works.
>
> We would like to emphasize that our work focuses on developing a variational denoising framework to understand the self-attention of transformers as a gradient descent approximation of a functional. From this new finding, we explain the oversmoothing issue of transformers as a result of self-attention minimizing a functional. **It is important to note that we primarily aim at explaining and resolving the collapse of representations, i.e., oversmoothing, at the token level (tokens representations become identical) rather than at the sentence level (the model gives high similarity scores for semantically different sentences)**. Although the merits of our method can be generalized to enhance sentence-level embeddings, it is out of the scope of our work, and we will explore it in future work.
>
> In addition, our new empirical study shows that SimCSE, proposed to enhance sentence representations via contrastive learning, still suffers from oversmoothing at the token level. Fig. 7 in the attached PDF shows that the cosine similarity between token representations for each layer of the SimCSE, which is trained on the STS-12 dataset, increases as the model depth grows. In contrast, our NeuTRENO is shown to mitigate the issue. When integrating NeuTRENO with the pre-trained SimCSE, without any fine-tuning, we observe the cosine similarity scores reduce significantly (Fig. 7)
>
> **References**
>
> [1] Gao, T., Yao, X., and Chen, D. "SimCSE: Simple Contrastive Learning of Sentence Embeddings". EMNLP, 2021.
>
> **Q4**. The paper did not investigate the transferability of the proposed approach.
>
> **Reply:** In our paper, we empirically study the transferability of NeuTRENO and summarize the results in Table 2 (Section 4). In particular, after pretraining DeiT and NeuTRENO DeiT on the ImageNet classification task, we finetune both models for the ADE20k image segmentation task. The results demonstrate that the finetuned NeuTRENO DeiT yields significantly better performance than the finetuned DeiT. Further details regarding this transfer learning experiment are provided in Section 4, from lines 238 to 244, and also in Appendix A2. Note that we provide the accuracies of the pre-trained models on the ImageNet classification task in Table 1.
>
> **Q5**. Some math symbols have not been or are vaguely defined, for example, D_x , D_{qk}, k(x,y), and K(x, y).
>
> **Reply:** Thank you for your comment. In our paper, $D_{x}$ represents the feature dimension of the token $x_{i}$, $i=1,\dots,N$, i.e. $X \in \mathbb{R}^{N \times D_{x}}$. $D_{qk}$ represents the feature dimension of $q_{i}$ and $k_{i}$, $i=1,\dots,N$, i.e., $Q, K \in \mathbb{R}^{N \times D_{qk}}$.
>
> The function $k(x, y)$ captures the similarity between signal values at positions x and y. For example, $k(x, y)$ can be a radial basis function (RBF) kernel $e^{-\gamma||\Phi(x) - \Phi(y)||^2}$ for a feature function $\Phi$ [1]. $K(x, y)$ is indeed just a shorthand notation for $k(x, y) + k(y, x)$. We use this notation to simplify the expression.
>
> **References**
>
> [1] Bishop, Christopher M. "Pattern Recognition and Machine Learning". New York. Springer, 2006.
>
>  **Q6**. The word “functional” is an adjective, so using it as a noun in the paper is ungrammatical.
>
> **Reply:** In the context of variational calculus, a functional is a mathematical concept that maps from a function space to the real numbers. A more formal definition can be found in [1].
>
> **References**
>
> [1] A. N. Kolmogorov and S. V. Fomin, “Elements of the Theory of Functions and Functional Analysis”, Graylock Press, 1957.
>
> -----
> We hope we have cleared your concerns about our work. We would appreciate it if we could get your further feedback at your earliest convenience.

---

> > ### Author Response · Authors · 2023-08-22
> > **Thank You**
> >
> > Dear Reviewer qvvE,
> >
> > Thanks for your reviews of our paper. Since the discussion period between the authors and the reviewers was already over and we have not heard from you during this period, we would be grateful if the reviewer could let us know if all your questions are addressed to some extent. If you are satisfied with our answers, we hope that the reviewer will consider adjusting your score.
> >
> > Best regards,
> >
> > Authors

---

### Author Rebuttal · Authors · 2023-08-09

Dear AC and reviewers,

Thanks for your thoughtful reviews and valuable comments, which have helped us improve the paper significantly. We are encouraged by the endorsements that: 1) Our paper's variational denoising framework for self-attention is novel and interesting  (Reviewer qvvE, h3ak, qcqa); 2) The derivation of the oversmoothing in transformers from our variational denoising framework is useful (Reviewer YNWk) and contributes to the understanding of the (oversmoothing) issue (Reviewer h3ak); 3) The theory in our paper is rigorous, in-depth (Reviewer k2Mg), clear, and detailed (Reviewer qvvE); 4) The effectiveness of our NeuTRENO model is demonstrated both theoretically and empirically (Reviewer h3ak).

Among the concerns of the reviewers is the significance of our NeuTRENO's advantages over the baseline transformer model. We address this concern here.

1. We provide the standard deviations from 5 runs for each experiment (in the main text) in Tables 9, 10, and 11 in the attached PDF. The standard deviations for both the NeuTRENO and baseline models in these tasks are small compared to the gains achieved by the NeuTRENO method. This observation suggests that NeuTRENO’s improvements over the baseline are significant and not by chance.

2. In addition to the standard metrics, we have evaluated the robustness of our NeuTRENO model compared to the baseline transformer model, particularly under adversarial examples and for out-of-distribution generalization. Table 12 in the attached PDF demonstrates that NeuTRENO DeiT-Tiny is consistently more robust than the DeiT-Tiny baseline on the Imagenet-C (common data corruption and perturbations, such as adding noise and blurring the images) [1], Imagenet-A (adversarial examples) [2], and Imagenet-R (out of distribution generalization) [3] datasets, which are widely used to test the model’s robustness.

3. Furthermore, in an incremental learning setting [4], our 8-layer NeuTRENO achieves 1.97% higher accuracy on the sentiment classification task [5] than the 8-layer baseline transformer.

4. To demonstrate the scalability of our proposed model, we have conducted additional experiments to show that our NeuTRENO method can effectively mitigate the oversmoothing issue in the BERT-base model. In particular, in Figure 7 (Left) in the attached PDF, we plot the cosine similarity between token representations across layers of a pre-trained BERT-base model [6] on the SQuAD v1.1 question answering task [7] and observe the presence of the oversmoothing issue as the model gets deeper, causing tokens to become identical. We then apply NeuTRENO on the same pre-trained BERT model, and without any fine-tuning, we observe a significant reduction in the cosine similarity between token embeddings in each layer (see Figure 7 (Left) in the attached PDF), indicating that NeuTRENO effectively mitigates the oversmoothing problem in BERT. Moreover, we have conducted the same analysis for a randomized BERT-base model and a randomized NeuTRENO BERT-base model and obtained the same encouraging results (see Figure 7 (Middle) in the attached PDF). These results further suggest that NeuTRENO helps alleviate the oversmoothing issue in large-scale transformer models.

We would also like to summarize the main contributions of our paper here. Our work first focuses on **developing a variational denoising framework to understand the self-attention of transformers as a gradient descent approximation of a functional**. Using this new finding, we **provide an explanation for the oversmoothing issue in transformers** as a result of self-attention minimizing a functional, leading to the smoothing effect on the input sequence, analogous to a diffusion process  (see Remark 1 in our main text). Then, we **rigorously prove this observation on the existence of oversmoothing in transformers using a random walk analysis** in Section 2.2. Thus, in our paper, we not only discuss the cause of oversmoothing but also prove its existence theoretically. Finally, we **propose the Neural Transformer with a Regularized Nonlocal Functional (NeuTRENO), a novel class of transformers designed to mitigate oversmoothing**. NeuTRENO is derived by optimizing a regularized nonlocal functional, which includes an additional convex fidelity term. This fidelity term penalizes the norm of the difference between the smooth output tokens from self-attention and the input tokens, thereby reducing the over-smoothing effect. It is important to note that in our analysis, we primarily aim at explaining and resolving the collapse of representations, i.e., oversmoothing, at the token level (tokens representations become identical) rather than at the sentence level (the models give high similarity scores for semantically different sentences).

**References**

[1] Hendrycks, Dan, et al. "Benchmarking neural network robustness to common corruptions and perturbations." arXiv, 2019.

[2] Hendrycks, Dan, et al. "Natural adversarial examples." CVPR, 2021.

[3] Hendrycks, Dan, et al. "The many faces of robustness: A critical analysis of out-of-distribution generalization." ICCV, 2021.

[4] Kahardipraja, et al. "Towards incremental transformers: An empirical analysis of transformer models for incremental NLU."  EMNLP, 2021.

[5] Dimitrios Kotzias, et al. "From group to individual labels using deep features". KDD, 2015

[6] Jacob Devlin, et. al. “BERT: Pre-training of Deep Bidirectional Transformers for Language Understanding”. NAACL, 2019.

[7] Pranav Rajpurkar, et al. “SQuAD: 100,000+ Questions for Machine Comprehension of Text”. EMNLP, 2016.

-----

We are glad to answer any further questions you have on our submission.

---

### Comment · Area_Chair_n6M1 · 2023-08-18
**Please read rebuttals**

Dear reviewers, if you didn't already, please read the rebuttals ASAP and at least acknowledge them explicitly.

Best,
Area Chair

---

### Author Response · Authors · 2023-08-19
**Additional Results on Oversmoothing and Empirical Performance**

Dear reviewers,

We would like to thank all reviewers again for your thoughtful reviews and valuable feedback. We have obtained additional empirical results to clarify the impact of oversmoothing on the model’s performance. We summarize our results below.

In Figure 7 (Left) in the rebuttal PDF, our NeuTRENO BERT finetuned on the SQUAD question answering task [1] yields better accuracy than the baseline BERT finetuned on the same task (81.39 exact match score and 88.62 F1 score vs. 80.77 exact match score and 88.12 F1 score). In Figure 7 (Right) in the rebuttal PDF, our NeuTRENO SimCSE, after finetuned on the STS-12 semantic textual similarity task [2], gains a significant improvement over the baseline SimCSE [3], which is also finetuned on the same task (77.32% vs. 75.29% Spearman's correlation. Here, the higher correlation, the better). These results indicate that reducing cosine similarity between tokens in the trained transformer-based models leads to better empirical performance.

We would also like to provide additional evidence that oversmoothing is indeed an issue for vision transformers (ViT). Indeed, the oversmoothing in ViT has been verified and investigated in existing works. In particular, [4] observes that the performance of ViT quickly saturates as more layers are added to the model. Moreover, experiments in [4] show that the 32-layer ViT underperforms the 24-layer ViT, indicating the difficulty of ViTs in gaining benefits from deeper architectures. The authors point out that oversmoothing results in this phenomenon by causing the token representations to become identical when the model grows deeper. Based on this observation, they propose a cross-head communication method that helps enhance the diversity of both token representations and attention matrices.

Furthermore, it has been shown in [5] that the training of ViT models encounters instability with greater depths. [6] proposes that this instability arises from the oversmoothing, where token representations for patches within an image become progressively alike as the model's depth increases. In an effort to explain this issue, [7] finds out that self-attention acts as a low-pass filter, which smoothens the token representations in ViTs. This leads to the proposal of the FeatScale method [7], which regulates feature frequencies, whether low or high, to counteract the consequences of oversmoothing.

Different from existing works, in our paper, we theoretically prove the oversmoothing phenomenon via the variational denoising framework that we develop. Our proposed NeuTRENO method helps mitigate the oversmoothing issue and improves the performance of the ViT baselines (DeiTs), as well as other baseline transformer-based models. We also empirically demonstrate that NeuTRENO not only reduces the cosine similarity between token representation (See Figure 1 in the main text and Figure 7) but also the redundancy in attention maps between layers (See Figure 5 in Appendix G2). Finally, we show that NeuTRENO is complementary to existing methods, including FeatScale [7] (See Table 1 and Table 7 in our paper).

We would be happy to do any follow-up discussion or address any additional comments.

**References**

[1] Pranav Rajpurkar, Jian Zhang, Konstantin Lopyrev, and Percy Liang. “SQuAD: 100,000+ questions for machine comprehension of text”. EMNLP, 2016.

[2] Eneko Agirre, et al. “SemEval-2012 task 6: A pilot on semantic textual similarity.” SemEval, 2012.

[3] Gao, T., Yao, X., and Chen, D. "SimCSE: Simple contrastive learning of sentence embeddings". EMNLP, 2021.

[4] Daquan Zhou, et al. “Deepvit: Towards deeper vision transformer”. arXiv, 2021.

[5] Touvron Hugo, et al. "Going deeper with image transformers." ICCV, 2021.

[6] Chengyue Gong, et al. “Vision transformers with patch diversification”. arXiv, 2021.

[7] Wang Peihao, et al. "Anti-oversmoothing in deep vision transformers via the fourier domain analysis: From theory to practice." ICLR, 2022.

---

### Decision · Program_Chairs · 2023-09-21

**Decision:**

Accept (poster)

**Comment:**

This paper studies how self-attention minimizes a functional which promotes smoothness, which had an effect on deeper transformer networks leading to token uniformity. The paper also proposes a novel regularizer term (NeuTRENO), based on the proportion of the difference between the first and current layers' self-attention activations. The experimental validation shows improvements (with NeuTRENO) compared to benchmark/standard very competitive transformers on ImageNet, ADE20k, WikiText103. Overall, all the reviewers who participated in the rebuttal agree that this paper is NeurIPS material. The rebuttal discussion/answers improved the clarity of the paper, which we hope to see in the camera ready.